# EA3D: Event-Augmented 3D Diffusion for Generalizable Novel View Synthesis

**Wangbo Yu**[1,2]* **Chaoran Feng**[1]* **Jianing Li**[3] **Aofan Zhang**[4] **Zhenyu Tang**[1]
**Mingyi Guo**[1] **Wei Zhang**[2] **Zhengyu Ma**[2] **Li Yuan**[1,2]† **Yonghong Tian**[2,1]†
[1]Shenzhen Graduate School, Peking University
[2]Peng Cheng Laboratory
[3]School of Computer and Science, Peking University
[4]Dalian University of Technology

## Abstract

We introduce **EA3D**, an Event-Augmented 3D Diffusion framework for *generalizable* novel view synthesis from event streams and sparse RGB inputs. Existing approaches either rely solely on RGB frames for generalizable synthesis, which limits their robustness under rapid camera motion, or require per-scene optimization to exploit event data, undermining scalability. **EA3D** addresses these limitations by jointly leveraging the complementary strengths of asynchronous events and RGB imagery. At its core lies a learnable *EA-Renderer*, which constructs view-dependent 3D features within target camera frustums by fusing appearance cues from RGB frames with geometric structure extracted from adaptively sliced event voxels. These features condition a *3D-informed diffusion model*, enabling high-fidelity and temporally consistent novel view generation along arbitrary camera trajectories. To further enhance scalability and generalization, we develop the *Event-DL3DV* dataset, a large-scale 3D benchmark pairing diverse synthetic event streams with photorealistic multi-view RGB images and depth maps. Extensive experiments on both real-world and synthetic event data demonstrate that **EA3D** consistently outperforms optimization-based and generalizable baselines, achieving superior fidelity and cross-scene generalization.

## 1 Introduction

Novel view synthesis and 3D scene reconstruction are fundamental tasks in computer vision, with broad applications in robotics (Rosinol et al., 2023; Zhu et al., 2022; Yen-Chen et al., 2021), autonomous driving (Yan et al., 2024; Lindström et al., 2024; Chen et al., 2025b), scene understanding (Kerr et al., 2023; Liu et al., 2023a; 2024b), and beyond. Recent advances in Neural Radiance Fields (NeRFs) (Mildenhall et al., 2020) and 3D Gaussian Splatting (3DGS) (Kerbl et al., 2023) have substantially improved the photorealism of novel-view rendering by learning dense and continuous scene representations. Despite their success, these approaches often struggle under high-speed motion and fail to generalize to new scenarios, primarily due to their dependence on densely sampled RGB frames and the need for per-scene optimization.

In novel view synthesis under fast camera motion, two major challenges undermine the performance of traditional NeRF- and 3DGS-based methods. First, rapid motion often limits the number of available training views, leading to an under-constrained reconstruction problem and causing overfitting to training views or convergence to trivial solutions. Second, large inter-frame distance caused by fast camera motion violates the smooth motion assumptions underlying feature matching, often resulting in unreliable initialization of camera poses in SfM pipelines (Schonberger & Frahm, 2016), which subsequently affects the optimization of NeRF (Mildenhall et al., 2020) or 3DGS (Kerbl et al., 2023). To address these challenges, recent works (Klenk et al., 2023; Rudnev et al., 2023; Xiong et al., 2024; Han et al., 2024; Cannici & Scaramuzza, 2024a; Qi et al., 2023; Yu et al., 2024a; Feng et al., 2025; Low & Lee, 2023; Cannici & Scaramuzza, 2024b; Bhattacharya et al., 2024; Hwang

---

*Equal contribution.
†Joint corresponding authors.

et al., 2023; Zahid et al., 2025; Yin et al., 2024; Liao et al., 2024) have explored the use of *event cameras* for novel view synthesis. Event streams captured by event cameras provide temporally dense, low-latency geometric cues that remain robust under fast motion and challenging lighting conditions, making them highly complementary to conventional frame-based cameras. In particular, several methods (Liao et al., 2024; Klenk et al., 2023; Xiong et al., 2024) demonstrate that fusing sparse RGB frames with continuous event streams enables accurate 3D reconstruction and novel view synthesis under fast camera motion. However, these approaches still rely on optimization-based 3D representations (Mildenhall et al., 2020; Kerbl et al., 2023), limiting their generalization to unseen environments. On the other hand, generalizable novel view synthesis methods (Yu et al., 2021; Xu et al., 2024b; Rockwell et al., 2021; Chen et al., 2024b; Liu et al., 2024a; Yu et al., 2025c; Jin et al., 2024) learn strong priors over 3D structure and appearance from large-scale multi-view datasets (Ling et al., 2024; Zhou et al., 2018; Reizenstein et al., 2021b; Yu et al., 2023b). However, their performance often deteriorates in the presence of wide-baseline input views.

To address these challenges, we propose **EA3D**, an Event-Augmented 3D Diffusion model for generalizable novel view synthesis from sparse RGB frames and continuous event streams. Our model consists of two main components: Firstly, drawing inspiration from cost volume-based novel view synthesis methods (Xu et al., 2024b; Chen et al., 2021; Liu et al., 2024c), we learn an Event-Augmented Feature Renderer (*EA-Renderer*) to generate 3D features for each target view along the novel trajectory, projecting both the appearance information from the RGB frames and the occlusion-resilient geometry information from the unposed event streams into target camera frustums. Secondly, we train a 3D-informed diffusion model conditioned on the 3D features, iteratively decoding these 3D features into consistent and photorealistic novel views. To support large-scale training and encourage strong generalization ability of our model, we introduce the *Event-DL3DV* dataset, a large-scale multi-view dataset consisting of real-world novel view sequences (Ling et al., 2024), diverse simulated event streams with randomized contrast thresholds, and per-view depth maps. Our model is trained end-to-end on the curated dataset and generalizes well to real-world event streams without requiring per-scene optimization.

Our main contributions are summarized as follows:

- We propose **EA3D**, the first generalizable framework for high-fidelity novel view synthesis from event streams and sparse RGB frames. To enable large-scale training and improve generalization across diverse scenes, we also introduce *Event-DL3DV*, a large-scale 3D dataset that pairs synthetic events with photorealistic multi-view RGB images and depth maps.

- We conduct extensive evaluations on both real-world event data and in-the-wild scenes, showing that EA3D consistently outperforms optimization-based and generalizable baselines, and demonstrates strong generalization across diverse scenarios.

## 2 RELATED WORKS

### 2.1 EVENT CAMERAS

Event cameras are bio-inspired sensors that offer high dynamic range and microsecond-level temporal resolution, making them well-suited for computer vision tasks in challenging conditions such as fast motion (Gallego et al., 2020; Brandli et al., 2014; Messikommer et al., 2025; Lin et al., 2023; Weng et al., 2021; Wan et al., 2025; Tulyakov et al., 2021a; Bardow et al., 2016; Pan et al., 2020; Zhu et al., 2019; Sun et al., 2022; Xu et al., 2024c). In novel view synthesis, recent studies demonstrate the effectiveness of event streams to improve performance under rapid motion (Klenk et al., 2023; Rudnev et al., 2023; Xiong et al., 2024; Han et al., 2024; Qi et al., 2023; Yu et al., 2024a; Feng et al., 2025). Notably, E-NeRF (Klenk et al., 2023), EF-3DGS (Liao et al., 2024) and Event3DGS (Xiong et al., 2024) achieve promising results by leveraging event streams and RGB frames for photorealistic novel view synthesis under fast camera motion. In addition, several recent works integrate event streams with 3DGS or related 3D reconstruction pipelines to address real-time rendering and motion blur. EventSplat incorporates event data into 3DGS to achieve real-time rendering from fast-moving event cameras (Yura et al., 2025). E2GS uses event streams to enhance 3DGS in the presence of motion blur and challenging illumination (Deguchi et al., 2024). DiET-GS combines events with diffusion priors to reconstruct sharp 3DGS scenes from heavily motion-blurred images (Lee & Lee, 2025),

while DeblurSplat employs events in an SfM-free pipeline for robust 3DGS-based deblurring (Li et al., 2025). EGS-SLAM leverages events to improve the robustness of RGB-D Gaussian Splatting SLAM under fast motion (Chen et al., 2025a), and E$^3$ NeRF exploits event streams to build efficient NeRFs from blurry images (Qi et al., 2024). However, these approaches still rely on optimizing a separate scene-specific representation and are not designed to serve as a general, training-free generative prior for event-augmented novel view synthesis across diverse scenes. In parallel, event cameras have advanced video frame interpolation (VFI), especially under large motion and motion blur where traditional RGB-based methods struggle. By capturing fine-grained motion cues, event streams help infer intermediate frames based on sparse frames more accurately (Tulyakov et al., 2021b; Paikin et al., 2021; He et al., 2022; Weng et al., 2023; Kim et al., 2023; Ma et al., 2024). Recently, generative approaches have further enhanced realism and generalization. EGVD (Zhang et al., 2025) introduced an event-guided video diffusion model for handling large motions, while others (Chen et al., 2024a) repurposed pretrained diffusion models for event-based interpolation. These methods focus on interpolating frames strictly along the event camera trajectory, and lack the ability to synthesize novel views from unseen views. Compared with above methods, our method enables high-quality novel view synthesis along flexible camera trajectories, and demonstrates strong generalization across diverse scenes.

## 2.2 Diffusion Model-based Novel View Synthesis

Diffusion models (Ho et al., 2020; Song et al., 2021; Rombach et al., 2022b) have shown strong potential for novel view synthesis from sparse inputs. GeNVS (Chan et al., 2023) and Zero-1-to-3 (Liu et al., 2023b) learn pose-conditioned diffusion models on large-scale datasets (Reizenstein et al., 2021a; Deitke et al., 2023; Chang et al., 2015), but are limited to specific categories (Watson et al., 2023) or synthetic scenes. While ZeroNVS (Sargent et al., 2024) and ReconFusion (Wu et al., 2024b) improve generation diversity, they are built on image diffusion models that synthesize each frame independently without explicitly modeling inter-frame dependencies, and therefore cannot enforce temporal consistency especially under large camera motions. Other works (Zhang et al., 2024; Chung et al., 2023; Shriram et al., 2024; Tang et al., 2025) refine warped depth-based views using pre-trained T2I diffusion models (Rombach et al., 2022a), often introducing artifacts in the inpainted region. More recently, video diffusion models have been explored for consistent novel view synthesis (Wang et al., 2024; Xu et al., 2024a; He et al., 2024; Sun et al., 2024). Several works employ point-based representations to guide novel view synthesis (You et al., 2025; Yu et al., 2025c;a). Others employ ray-map–conditioned video diffusion pipelines (He et al., 2024; Xu et al., 2024a; Gao et al., 2024; Wu et al., 2024a; Yu et al., 2023a; 2024b; Zhou et al., 2025; Yu et al., 2025b; Ma et al., 2025). Although these models learn strong priors from large-scale multi-view datasets, their inability to leverage event data limits performance in challenging fast-motion and motion-blur scenarios.

## 3 Method

As shown in Fig. 1, EA3D consists of two key components. First, given a novel view camera trajectory, an Event-Augmented Feature Renderer (*EA-Renderer*) projects the continuous event streams and the sparse RGB frames into 3D features for each target camera frustum. Second, a *3D-aware diffusion model* takes the resulting 3D features as input and synthesizes photorealistic novel views. Below we describe the core components of our method. More details about model architecture and dataset curation are provided in Appendix A.

### 3.1 Event-Augmented Feature Renderer

The event stream captured during fast camera motion provides dense and temporally continuous geometric prior of the 3D scene. However, event stream inherently lacks appearance information such as color and texture. In contrast, conventional RGB frames contain rich appearance content but offer only sparse and incomplete geometric cues, especially in the presence of fast camera motion. To synthesize photorealistic novel views, we propose a learnable EA-Renderer that unifies both modalities into a consistent, 3D-aware feature representation. Without loss of generality, taking novel view synthesis from two RGB frames $(\mathbf{I}_{t_0}, \mathbf{I}_{t_1})$ and the continuous event stream $\mathbf{E}(t_0, t_1)$ captured in between as an example, where the general multi-view case can be naturally decomposed into a set of two-view subproblems. Given a novel-view camera trajectory $\{\mathbf{T}^i\}_{i=1}^N$ between the two frames, the

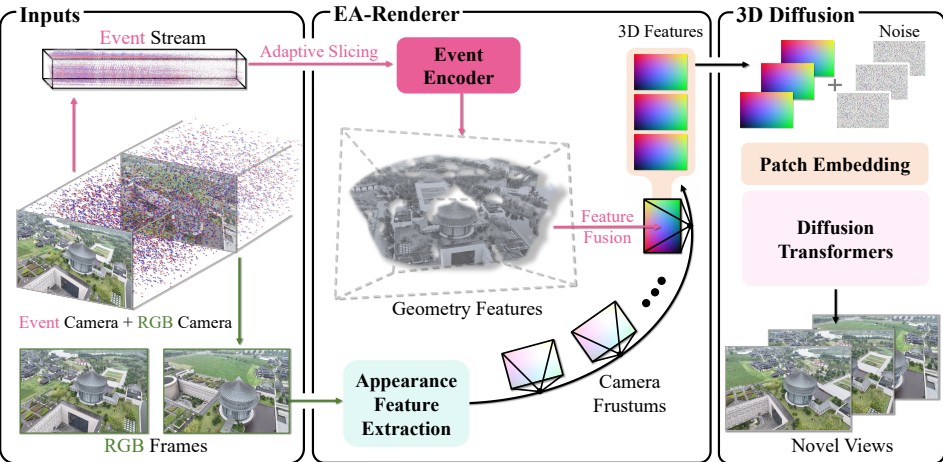

Figure 1: **Overview of EA3D.** Given a set of sparse RGB frames and continuous event streams, we learn an Event-Augmented Feature Renderer (*EA-Renderer*) to construct view-dependent 3D features by projecting both appearance cues from RGB frames and occlusion-resilient geometry features from adaptively sliced event voxel grids into each target camera frustum. These 3D features are then passed into a conditional video diffusion model as 3D conditions, facilitating photorealistic and consistent novel view synthesis.

EA-Renderer renders along a camera trajectory and produces a sequence of 3D features $\mathbf{F}_{3D}$ aligned with the target camera frustums. The EA-Renderer is structured in three stages: Appearance Feature Extraction, Event Feature Extraction and Feature Fusion.

**Appearance Feature Extraction**    Given a pair of RGB frames $(\mathbf{I}_{t_0}, \mathbf{I}_{t_1})$, we first obtain their camera parameters and depths using an off-the-shelf multi-view stereo model (Wang et al., 2025). These RGB frames are then projected into each camera frustum of the novel view trajectory $\{\mathbf{T}^i\}_{i=1}^{N}$, producing a sequence of view projections $\{\mathbf{P}^i\}_{i=1}^{N}$, which are then passed through an appearance encoder $\mathcal{E}_{\text{appr}}$, resulting view-wise appearance feature maps:

$$\{\mathbf{F}_{\text{appr}}^i\}_{i=1}^{N} = \mathcal{E}_{\text{appr}}(\{\mathbf{P}^i\}_{i=1}^{N}). \tag{1}$$

These appearance features contain rich texture information. However, due to large view baselines and occlusions between the RGB frames, they fail to capture the complete scene geometry. Therefore, we further introduce geometry cues derived from the event streams.

**Event Feature Extraction**    The continuous event stream offers microsecond-level latency and an extremely high temporal resolution, which provides temporally dense, occlusion-resilient observations of the 3D scene. To extract geometric information from the event stream, we adopt a voxel grid-based event representation (Gallego et al., 2020) and leverage an adaptive slicing strategy to obtain event voxel grids. Specifically, we first partition the continuous event stream $\mathbf{E}(t_0, t_1)$ into $N$ temporal segments. For each segment, we construct two temporally overlapping slices: a short slice containing $m$ events to preserve short-term scene information, and a long slice containing $2m$ events to capture longer temporal context. To ensure sufficient voxel density under the non-uniform event stream, the time duration of each slice is adaptively adjusted until the required number of events is accumulated. The resulting $N$ short slices and $N$ long slices are then combined along the channel dimension, yielding a temporally enriched event voxel grid $\{\mathbf{E}^i\}_{i=1}^{N}$. We then learn an event encoder $\mathcal{E}_{\text{event}}$ to project the event voxel grid into event features:

$$\mathbf{F}_{\text{event}} = \mathcal{E}_{\text{event}}(\{\mathbf{E}^i\}_{i=1}^{N}). \tag{2}$$

The resulting event features $\mathbf{F}_{\text{event}}$ encode structural continuity and occlusion-resilient geometry. However, since obtaining accurate poses and depths for event streams is non-trivial, directly projecting them into the target camera frustum along the novel view trajectory is challenging. In addition, event features inherently lack appearance cues such as color and texture. To address these issues, we introduce a feature fusion module to integrate the event features with appearance features.

**Feature Fusion**   We learn a cross-attention layer to fuse the un-posed event features with the posed appearance features. For each appearance feature $\mathbf{F}_{\text{appr}}^i$ in the novel view trajectory, we map $\mathbf{F}_{\text{appr}}^i$ into query matrix and map the entire event features $\mathbf{F}_{\text{event}}$ into key and value matrices, then computing attention map to obtain the final 3D features:

$$\{\mathbf{F}_{3D}\}_{i=1}^N = \{\text{Attention}(Q(\mathbf{F}_{\text{appr}}^i), K(\mathbf{F}_{\text{event}}), V(\mathbf{F}_{\text{event}}))\}_{i=1}^N. \tag{3}$$

The resulting $\{\mathbf{F}_{3D}\}_{i=1}^N$ project occlusion-resilient geometry priors and appearance information into each target camera frustums of the novel view trajectory, serving as strong 3D prior to guide the diffusion model to generate high-quality novel views.

## 3.2   3D-AWARE DIFFUSION MODEL

As shown in Fig. 1, given a novel-view camera trajectory $\{\mathbf{T}^i\}_{i=1}^N$, we first render the 3D features $\mathbf{F}_{3D}$ using our EA-Renderer. We then learn a conditional distribution $\mathbf{I} \sim p(\mathbf{I} \mid \mathbf{F}_{3D})$ to map these features into high-quality novel-view $\{\mathbf{I}^i\}_{i=1}^N$. To encourage 3D consistency in novel view synthesis, we model the conditional distribution using a video diffusion model conditioned on $\mathbf{F}_{3D}$. Our implementation builds upon the open-sourced image-to-video generation variant of CogVideoX (Yang et al., 2024), which adopts Diffusion Transformers with 3D self-attention for spatio-temporally coherent image-to-video generation, making it highly suitable for our setting.

Originally, CogVideoX is designed to take a single input image of shape $H \times W \times 3$ and generate a video of shape $N \times H \times W \times 3$. To match the temporal length of the target video, the input image is temporally padded to construct a condition video of shape $N \times H \times W \times 3$, which is then passed through the 3D VAE encoder of CogVideoX to obtain a condition feature with a shape of $\frac{N}{4} \times \frac{H}{8} \times \frac{W}{8} \times C$. This condition feature is concatenated with sampled Gaussian noise and transformed into tokens via a patch embedding layer. The tokens are then iteratively refined by Diffusion Transformer (DiT) blocks (Peebles & Xie, 2023) during the denoising process. Finally, the clean latent is unpatchified and decoded by the VAE decoder to reconstruct the output video.

To adapt CogVideoX (Yang et al., 2024) for the novel view synthesis task, we replace its original image-based condition feature with 3D features $\mathbf{F}_{3D}$ rendered by our EA-Renderer. In this adaptation, we also repurpose the space-time VAE encoder from CogVideoX as our appearance encoder $\mathcal{E}_{\text{appr}}$ to reduce the domain gap and facilitate convergence during training. Since the 3D VAE encoder of CogVideoX includes a temporal compression mechanism, the number of output appearance features $\mathbf{F}_{\text{appr}}$ exactly reduced to $\frac{N}{4}$. Consequently, the final output 3D feature $\mathbf{F}_{3D}$ has a shape of $\frac{N}{4} \times \frac{H}{8} \times \frac{W}{8} \times C$. It is then concatenated with Gaussian noise and patchified with a newly initialized patch embedding layer to form tokens matching the original input size of the DiT blocks. The noisy tokens are subsequently denoised by the DiT blocks, and finally unpatchified and decoded by the VAE decoder to reconstruct the novel views $\{\mathbf{I}^i\}_{i=1}^N$.

## 3.3   TRAINING DETAILS

The model is trained end-to-end using a combination of diffusion loss and a reconstruction loss, both weighted equally. For the diffusion loss, we adopt the standard noise schedule and loss formulation from CogVideoX (Yang et al., 2024), enabling compatibility with pretrained weights and stable convergence in the 3D-aware generation setting:

$$\mathcal{L}_{\text{diffusion}} = \mathbb{E}_{\mathbf{I}, \mathbf{F}_{3D}, t, \epsilon,}[\|\epsilon - \epsilon_\theta(\mathbf{I}, t, \mathbf{F}_{3D})\|_2^2]. \tag{4}$$

To further stabilize training and accelerate convergence, we impose a reconstruction loss between the 3D features $\mathbf{F}_{3D}$ rendered from EA-Renderer and the ground-truth novel view features $\mathcal{E}_{\text{appr}}(\mathbf{I})$ obtained from the 3D VAE encoder of CogVideoX (Yang et al., 2024):

$$\mathcal{L}_{\text{recon}} = \|\mathbf{F}_{3D} - \mathcal{E}_{\text{appr}}(\mathbf{I})\|_2^2. \tag{5}$$

During training, we jointly optimize the event encoder and the feature fusion module in the EA-Renderer, as well as the patch embedding layer and the DiT blocks in the video diffusion model. The training resolution is fixed at $384 \times 672$, with a novel view sequence length of 49 frames. The event stream is sliced by uniformly sampling $m \in [1 \times 10^5, \ 3 \times 10^5]$, which enhances robustness to event stream fluctuations. Training is conducted for 12,000 iterations using a mini-batch size of 8 across 8 GPUs (each with 80 GB of memory), with a learning rate set to $1 \times 10^{-5}$.

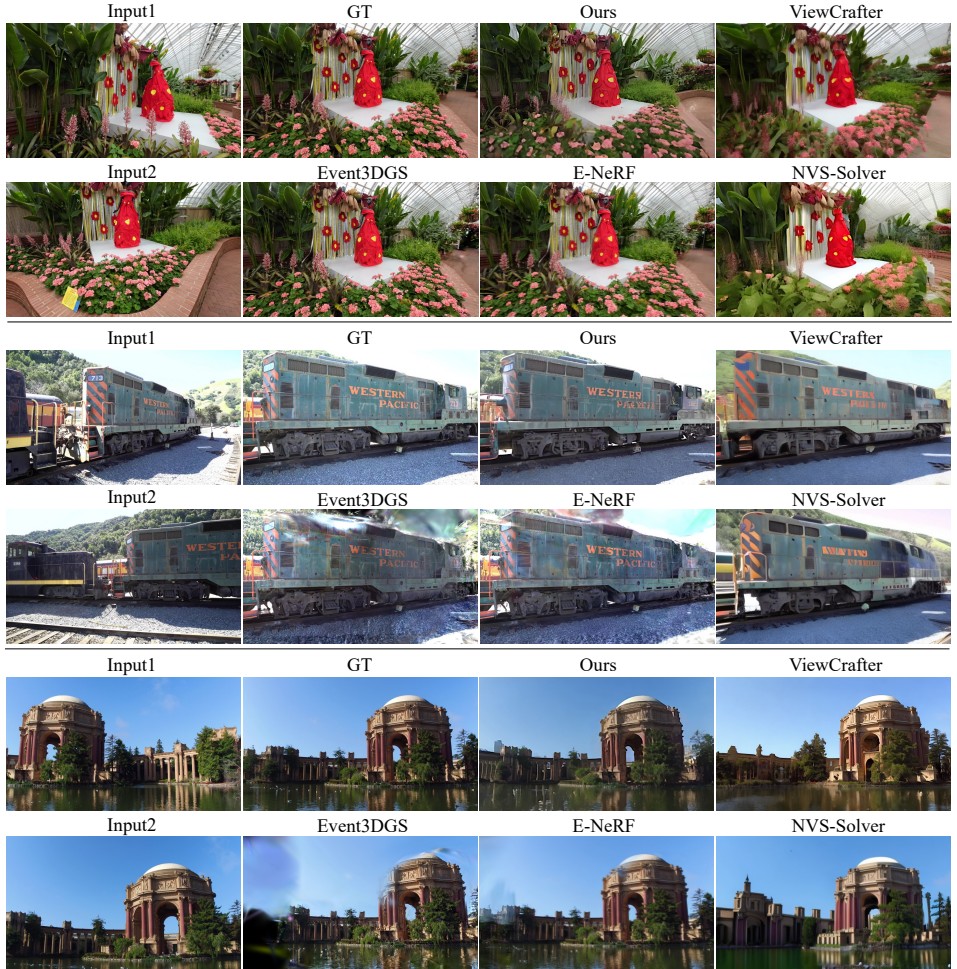

Figure 2: **Qualitative comparison on in-the-wild scenes.** We show results on the challenging 2-view input setting with large view baselines. ViewCrafter (Yu et al., 2025c) and NVS-Solver (You et al., 2025) exhibit visible artifacts and geometry degradation. In contrast, our method reconstructs sharper textures and more complete geometry by leveraging temporally dense geometric priors from events.

Table 1: **Quantitative comparison on in-the-wild scenes.** We evaluate our model on the DL3DV (Ling et al., 2024) and Tanks-and-Temples (Knapitsch et al., 2017) (T&T) benchmarks under 2, 4, and 6 input views.

|  | Method | 2 Views | | | 4 Views | | | 6 Views | | |
|---|---|---|---|---|---|---|---|---|---|---|
|  |  | PSNR ↑ | SSIM ↑ | LPIPS ↓ | PSNR ↑ | SSIM ↑ | LPIPS ↓ | PSNR ↑ | SSIM ↑ | LPIPS ↓ |
| *DL3DV* | E-NeRF (Klenk et al., 2023) | 18.01 | 0.627 | 0.314 | 22.97 | 0.720 | 0.233 | 25.19 | 0.778 | 0.212 |
|  | Event3DGS (Han et al., 2024) | 16.84 | 0.505 | 0.431 | 22.10 | 0.715 | 0.263 | 25.26 | 0.800 | 0.189 |
|  | ViewCrafter (Yu et al., 2025c) | 19.10 | 0.698 | 0.324 | 20.78 | 0.737 | 0.261 | 22.51 | 0.732 | 0.230 |
|  | NVS-Solver (You et al., 2025) | 17.75 | 0.633 | 0.340 | 21.83 | 0.702 | 0.277 | 22.18 | 0.725 | 0.241 |
|  | Ours | **22.82** | **0.732** | **0.251** | **24.80** | **0.793** | **0.186** | **25.41** | **0.830** | **0.166** |
| *T&T* | E-NeRF (Klenk et al., 2023) | 22.96 | 0.651 | 0.302 | 25.46 | 0.748 | 0.241 | 26.21 | 0.787 | 0.212 |
|  | Event3DGS (Han et al., 2024) | 22.42 | 0.632 | 0.319 | **25.54** | 0.754 | 0.237 | **26.32** | 0.791 | 0.206 |
|  | ViewCrafter (Yu et al., 2025c) | 18.24 | 0.607 | 0.289 | 22.26 | 0.754 | 0.251 | 22.87 | 0.793 | 0.213 |
|  | NVS-Solver (You et al., 2025) | 17.68 | 0.615 | 0.313 | 20.57 | 0.688 | 0.296 | 20.85 | 0.721 | 0.269 |
|  | Ours | **23.50** | **0.756** | **0.218** | 24.77 | **0.780** | **0.183** | 25.84 | **0.831** | **0.165** |

## 4 EXPERIMENTS

We begin by summarizing the evaluation setup in Sec. 4.1. Sec. 4.2 then reports quantitative and qualitative comparisons on the benchmarks. Sec. 4.3 presents ablation studies validating our model design and training losses. Appendix B.2 reports additional runtime and memory analysis, while Appendix B.3–B.7 present further ablations on robustness to trajectory misalignment, motion blur, contrast thresholds, and additional perceptual comparisons.

### 4.1 EXPERIMENTAL SETTING

**Comparison Baselines**   We compare our method against both optimization-based methods and RGB-only generalizable novel view synthesis approaches. For optimization-based baselines, we include Event3DGS (Han et al., 2024), which integrates asynchronous event streams into the 3DGS optimization. It can also be adapted to incorporate RGB frames as additional supervision for colored novel view synthesis. We also compare with E-NeRF (Klenk et al., 2023), which extends NeRF-based novel view synthesis to event cameras by reconstructing continuous radiance fields from temporally aggregated event streams. E-NeRF supports hybrid supervision and can generate photorealistic views by incorporating sparse RGB frames during training. For RGB-only generalizable baselines, we include NVS-Solver (You et al., 2025), a method that utilizes a video diffusion model (Blattmann et al., 2023) to inpaint depth-warped views for novel view synthesis. We also compare with ViewCrafter (Yu et al., 2025c), which integrates point-based 3D reconstruction with a video diffusion model, enabling novel view generation from sparse RGB inputs with explicit camera pose control.

**Evaluation Data**   We first evaluate our model on in-the-wild scenes to assess its generalization ability across diverse environments. Then, we conduct experiments on datasets containing real event data to verify the robustness of our model under real-world event inputs. For the in-the-wild scene comparison, we use 140 test scenes from the DL3DV benchmark (Ling et al., 2024) that do not overlap with our training data, as well as 10 scenes from the Tanks-and-Temples (T&T) (Knapitsch et al., 2017) dataset, for both qualitative and quantitative evaluation. For the real event data comparison, since there are no existing novel view synthesis benchmarks that include both sharp RGB frames and event data, we filter out 7 static sequences from the DSEC (Gehrig et al., 2021) dataset that contains both sharp RGB frames and real-captured event data for evaluation.

**Evaluation Setting**   Given a test novel view sequence from the evaluation dataset, we experiment with novel view synthesis under 2, 4, and 6 input views. For the optimization-based baselines E-NeRF (Klenk et al., 2023) and Event3DGS (Han et al., 2024), since they are designed to synthesize novel views along the event camera trajectory, we follow their original setting and simulate event streams directly from the ground truth novel view sequence using vid2e (Hu et al., 2021) to ensure their rendered novel views align with ground truth. In contrast, our method is designed to support novel view synthesis along flexible camera trajectories without requiring strict alignment to the event camera trajectory. To verify this capability, we sample sparse frames that do not overlap with the ground truth views from the test sequence, and use vid2e to simulate event streams based on these sampled frames. As a result, the simulated event stream used in our method is misaligned with the ground truth novel views, posing a more general and challenging setting. For fairness, all simulations are performed with the same contrast threshold range and event simulator configuration.

### 4.2 COMPARISON RESULTS

**In-the-wild Scene Comparison**   We evaluate our method on DL3DV (Ling et al., 2024) and Tanks-and-Temples (T&T) (Knapitsch et al., 2017), under 2, 4, and 6 input views. Table 1 reports the quantitative results. Compared to the optimization-based baselines (E-NeRF (Klenk et al., 2023) and Event3DGS (Han et al., 2024)), our generalizable model achieves the highest performance in the most challenging 2-view setting. For 4-view and 6-view inputs, our model achieves comparable or better results, demonstrating the strong generalization ability of our method across different scenes. Compared to generalizable RGB-only baselines (ViewCrafter (Yu et al., 2025c) and NVS-Solver (You et al., 2025)), our method consistently outperforms these methods across all settings, which confirms the effectiveness of our event-augmented design in enhancing generation fidelity under large viewpoint changes. For qualitative comparison, Fig. 2 visualizes representative synthesis results. It can be found that results of ViewCrafter (Yu et al., 2025c) and NVS-Solver (You et al.,

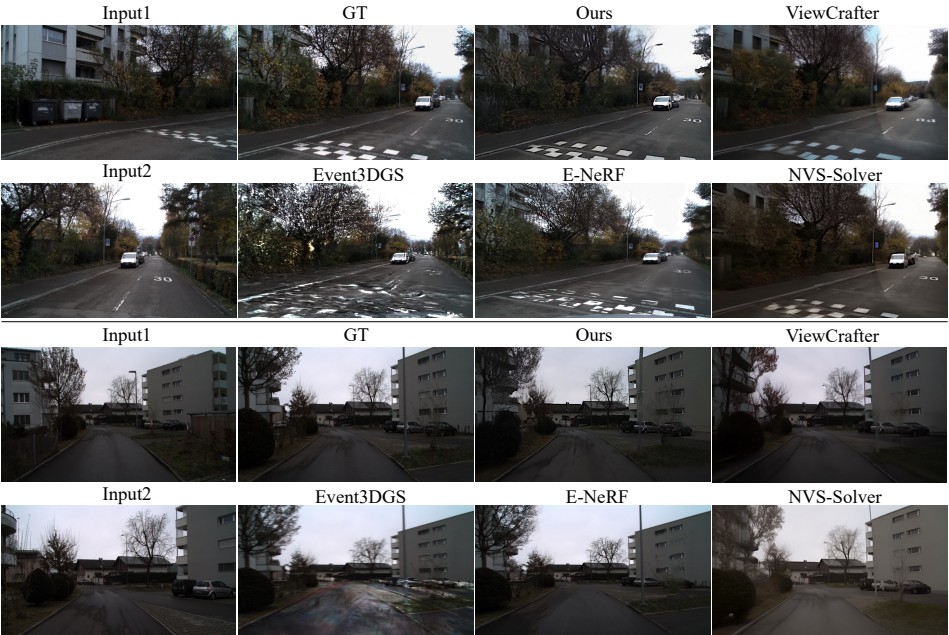

Figure 3: **Qualitative comparison on real event data.** Our method produces sharper textures and more complete geometry compared to both optimization-based and RGB-only baselines, demonstrating its robustness under real-world event inputs.

Table 2: **Quantitative comparison on real event data.** We report performance under 2, 4, and 6 input views on the DSEC dataset (Gehrig et al., 2021).

| Method | 2 Views | | | 4 Views | | | 6 Views | | |
|---|---|---|---|---|---|---|---|---|---|
| | PSNR ↑ | SSIM ↑ | LPIPS ↓ | PSNR ↑ | SSIM ↑ | LPIPS ↓ | PSNR ↑ | SSIM ↑ | LPIPS ↓ |
| E-NeRF (Klenk et al., 2023) | 15.52 | 0.622 | 0.503 | 22.03 | 0.763 | 0.369 | 23.25 | 0.816 | 0.345 |
| Event3DGS (Han et al., 2024) | 14.63 | 0.605 | 0.518 | 21.46 | 0.745 | 0.377 | 22.98 | 0.802 | 0.361 |
| ViewCrafter (Yu et al., 2025c) | 18.71 | 0.684 | 0.279 | 21.65 | 0.752 | 0.232 | 22.50 | 0.785 | 0.261 |
| NVS-Solver (You et al., 2025) | 18.68 | 0.689 | 0.283 | 21.49 | 0.736 | 0.247 | 21.61 | 0.777 | 0.252 |
| Ours | **24.89** | **0.792** | **0.211** | **26.31** | **0.827** | **0.195** | **26.87** | **0.835** | **0.177** |

2025) suffer from structural distortions and texture inconsistencies under large viewpoint changes. E-NeRF (Klenk et al., 2023) and Event3DGS (Han et al., 2024) also suffer from artifacts introduced by the optimization process. In comparison, our method produces more complete geometry and sharper textures, demonstrating the benefits of incorporating temporally dense event information into novel view synthesis.

**Real Event Data Comparison**   To evaluate the robustness of our method under real-world event streams, we conduct experiments on static driving scenes selected from the DSEC dataset (Gehrig et al., 2021), which provides synchronized event data and sharp RGB frames. Quantitative results under 2, 4, and 6 input views are summarized in Table 2. Our method achieves the best performance across all metrics and view settings. Qualitative results are visualized in Fig. 3, where our method generates more accurate spatial layouts compared to the baselines. These results highlight our model's ability to generalize from simulated to real event data.

### 4.3 ABLATION STUDY

To ablate our model design choices and training losses, we conduct a series of experiments on the Tanks-and-Temples (Knapitsch et al., 2017) dataset and real event data from the DSEC (Gehrig et al., 2021) dataset under the challenging 2-view input setting.

Table 3: **Quantitative ablation on model design and training loss.** Experiments are conducted under the challenging 2-view setting on the Tanks-and-Temples (Knapitsch et al., 2017) benchmark and real event data from the DSEC (Gehrig et al., 2021) dataset.

| Model Variant | T&T | | | Real Event Data | | |
|---|---|---|---|---|---|---|
| | PSNR ↑ | SSIM ↑ | LPIPS ↓ | PSNR ↑ | SSIM ↑ | LPIPS ↓ |
| w/o Geometry Feature | 18.87 | 0.631 | 0.285 | 18.90 | 0.672 | 0.275 |
| w/o Reconstruction Loss | 20.39 | 0.670 | 0.271 | 19.82 | 0.651 | 0.280 |
| w/o Adaptive Slicing | 22.96 | 0.724 | 0.235 | 23.06 | 0.778 | 0.248 |
| Ours | **23.50** | **0.756** | **0.218** | **24.85** | **0.789** | **0.215** |

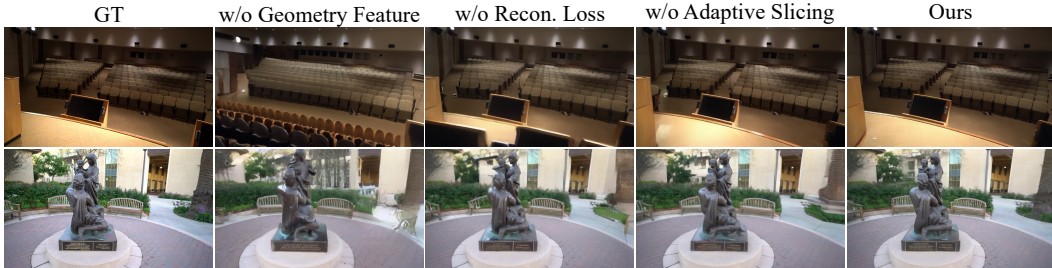

| GT | w/o Geometry Feature | w/o Recon. Loss | w/o Adaptive Slicing | Ours |

Figure 4: **Qualitative ablation on the model design and training loss.** Experiment conducted under the challenging 2-view setting.

**Effectiveness of Geometry Features from Event Streams** Firstly, to verify the importance of geometry features extracted from event streams, we train an ablated variant of our model by removing the event encoder and the feature fusion module, feeding only the appearance features into the diffusion model. As shown in Table 3 and Fig. 4, the performance drops significantly in the challenging 2-view setting, where the appearance features alone provide insufficient geometric cues to resolve occlusions and maintain structural consistency.

In addition, to validate the effectiveness of geometry features under large-baseline conditions where appearance features alone provide little to no visible overlap, we compare the performance change between our full model and the ablated variant without geometry features. Specifically, we define the view range as the number of frames between the two input views selected from the test sequence. To

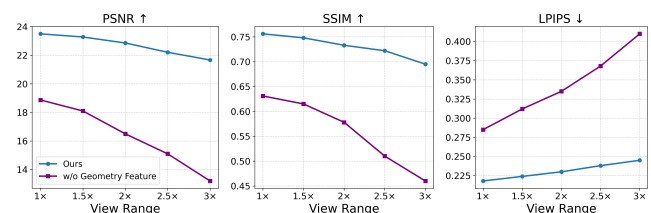

Figure 5: Ablation on the effectiveness of geometry features extracted from event streams under increasing view range.

simulate increasingly challenging conditions, we progressively enlarge this inter-frame distance by increasing the number of intermediate frames between the two inputs. Starting from the basic view range adopted in Table 3, we expand the test view range to $1.5\times$, $2\times$, $2.5\times$, and $3\times$ of the base view range, thereby creating scenarios with decreasing appearance overlap and greater geometric ambiguity. The results are visualized in Fig. 5, showing that by incorporating geometry features extracted from event streams, our model maintains more stable performance as the view range increases.

**Effectiveness of Adaptive Event Slicing** We evaluate the effectiveness of the adaptive slicing strategy in event feature extraction by comparing it with a naive fixed-duration slicing baseline. As shown in Table 3 and Fig. 4, incorporating adaptive slicing improves novel view synthesis quality and reduces artifacts in the generated results.

**Ablation on Reconstruction Loss** To assess the impact of the reconstruction loss $\mathcal{L}_{\text{recon}}$, we conduct an ablation study by training our model using only the diffusion loss $\mathcal{L}_{\text{diffusion}}$. As shown in Table 3 and Fig. 4, removing $\mathcal{L}_{\text{recon}}$ leads to noticeable degradation in both structural consistency and

perceptual quality. This demonstrates that explicitly supervising the EA-Renderer helps align the feature space and improve generation fidelity.

## 5  CONCLUSION

We presented **EA3D**, a generalizable framework that unifies event streams and sparse RGB frames for high-fidelity novel view synthesis. By introducing a learnable Event-Augmented Feature Renderer (*EA-Renderer*) and conditioning a *3D-informed diffusion model* on fused 3D features, our method effectively captures both geometric continuity from events and appearance richness from RGB inputs. To support large-scale learning, we constructed the *Event-DL3DV* dataset, which provides RGB images paired with simulated event streams and dense depth maps. Extensive experiments on both synthetic and real-world benchmarks demonstrate that EA3D outperforms existing optimization-based and generalizable methods, especially in challenging scenarios with large viewpoint baselines.

**Limitations**  As a diffusion-based framework, our method still faces limitations in inference efficiency. In addition, the model may encounter challenges when the input views are of extremely low quality, making the initial MVS step produce inaccurate camera poses or making our model difficult to extract appearance information across views.

## REPRODUCIBILITY STATEMENT

Implementation details for **EA3D** are provided in Sec. 3 and Appendix A. Upon publication, we will release the complete codebase and processed datasets to facilitate full reproducibility.

## ACKNOWLEDGMENTS

This work was supported in part by the Natural Science Foundation of China (No. 62332002, 62425101),The Guangdong Grants (Grant No.2023ZT10X075), and Shenzhen Science and Technology Program(KQTD202407291020051063).

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

This supplementary material provides expanded implementation details, additional experiments, and a broader impact discussion. Sections A, B, and C outline the relevant additions referenced in the main text.

# A  IMPLEMENTATION DETAILS

## A.1  MODEL ARCHITECTURE

**Event Stream Processing**  We adopt a voxel grid-based event representation (Gallego et al., 2020) and leverage an adaptive slicing strategy to obtain event voxel grids. As shown in Fig. 6, the continuous event stream $\mathbf{E}(t_0, t_1)$ is first partitioned into $N$ temporal segments. For each segment, we construct two temporally overlapping slices: a short slice containing $m$ events to preserve short-term scene information, and a long slice containing $2m$ events to capture longer temporal context. To ensure sufficient voxel density under the non-uniform event stream, the

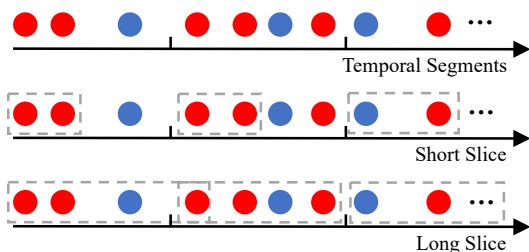

Figure 6: Illustration of adative event slicing.

time duration of each slice is adaptively adjusted until the required number of events is accumulated. The resulting $N$ short slices and $N$ long slices are then combined along the channel dimension to form a two-channel voxel input $\{\mathbf{E}^i\}_{i=1}^N$, where $\mathbf{E}_i \in \mathbb{R}^{H \times W \times 2}$.

**Event Encoder**  The encoder consists of four 3D convolutional blocks with increasing channels and spatial-temporal downsampling via kernel size $3 \times 3 \times 3$ and stride $2 \times 2 \times 2$, followed by group normalization and ReLU activation of each block. The output geometry feature is a compact representation $\mathbf{F}_{\text{event}}$ with shape $\frac{N}{8} \times \frac{H}{8} \times \frac{W}{8} \times C$. These features encode occlusion-resilient structure and are fused with appearance features through cross-attention for view-conditioned 3D feature generation. We provide a visualization of the learned event features in Fig. 8. It can be observed that the event features effectively capture the scene structure.

**Feature Fusion**  We adopt Perceiver cross-attention (Jaegle et al., 2021) to fuse the posed appearance features with the unposed event features for view-informed 3D feature generation. For each novel-view appearance feature $\mathbf{F}_{\text{appr}}^i \in \mathbb{R}^{\frac{H}{8} \times \frac{W}{8} \times C}$, we first flatten it into a sequence of $N_1 = \frac{H}{8} \cdot \frac{W}{8}$ tokens and apply a linear layer to project it into query matrix $Q$. The shared event feature volume $\mathbf{F}_{\text{event}} \in \mathbb{R}^{\frac{N}{8} \times \frac{H}{8} \times \frac{W}{8} \times C}$ is similarly reshaped into $N_2 = \frac{N}{8} \cdot \frac{H}{8} \cdot \frac{W}{8}$ tokens and linearly projected to $K, V$ matrices. We then apply perceiver cross-attention to obtain the final output 3D feature, and reshape it back to the original spatial size to obtain $\mathbf{F}_{3D}^i \in \mathbb{R}^{\frac{H}{8} \times \frac{W}{8} \times C}$. Finally, we concatenate the per-frame fused features along the temporal axis to obtain the full 3D feature $\mathbf{F}_{3D} \in \mathbb{R}^{\frac{N}{4} \times \frac{H}{8} \times \frac{W}{8} \times C}$, which serves as the input to the diffusion model for temporally coherent novel view synthesis.

## A.2  DATASET

We introduce *Event-DL3DV*, which augments the DL3DV dataset (Ling et al., 2024) with event streams and depths. Specifically, the original DL3DV dataset contains 10,000 diverse static 3D scenes with multi-view RGB images. For each view sequence in DL3DV, we simulate event streams using the event simulator vid2e (Hu et al., 2021). We use the multi-view stereo model VGGT (Wang et al., 2025) to compute per-frame depths and generate RGB projections along the estimated camera trajectory, which are used for appearance feature extraction. In total, we generate 10,000 sequences with event streams, ground-truth novel views and depths.

**Event Threshold Augmentation**  During event stream simulation, the event triggering is based on the change in log intensity at each pixel exceeding a contrast threshold, mimicking the behavior of real event cameras. Following the stochastic simulation strategy proposed in E2VID (Rebecq et al., 2019), we introduce randomness into the threshold selection by sampling positive and negative thresholds from a uniform distribution $\mathcal{U}(0.05, 0.3)$. This enables the generation of event streams with varying sensitivity levels and sparsity, enhancing the diversity of the training data.

**Resolution Augmentation**  To simulate different types of event cameras with varying spatial resolutions, we apply resolution-based augmentation to the input RGB frames before event generation.

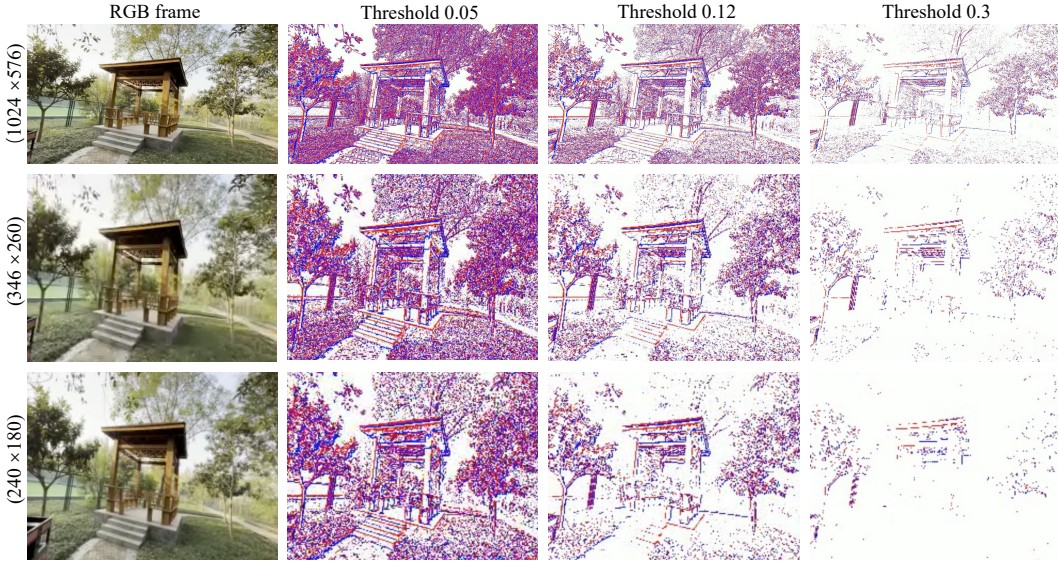

Figure 7: **Event simulation under different contrast thresholds and resolutions.** Each row corresponds to a simulated resolution: $1024 \times 576$, $346 \times 260$, and $240 \times 180$, respectively. Each column shows the simulated events under different contrast thresholds: 0.05, 0.12, and 0.3. Lower thresholds lead to denser event firing with more fine-grained structure, while higher thresholds produce sparser events primarily along strong edges. To improve robustness across varying event data quality and settings, we train our model with mixed simulated events from diverse thresholds and resolutions.

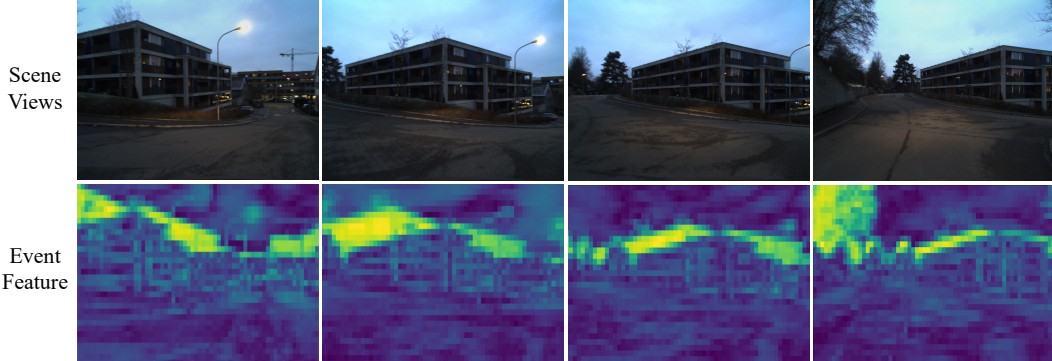

Figure 8: **Visualization of event features.** The top row shows the RGB images of the scene, while the bottom row shows the visualizations of the event features. The feature maps clearly reveal structural information such as object edges and contours, indicating that the event encoder successfully captures the geometric details of the scene.

Specifically, we resize the frames to different target resolutions prior to feeding them into the event simulator. This allows us to generate event streams that approximate the characteristics of real devices such as the DAVIS346 ($346 \times 260$) and DAVIS240 ($240 \times 180$). In addition, we simulate high-resolution event streams up to $1024 \times 576$. While event cameras with such high resolutions are not yet common, recent sensors such as the Sony IMX636 support resolutions up to $1280 \times 720$. We include such high-resolution streams to increase training data diversity and improve the robustness of the model. Examples are shown in Fig. 7.

**Blur Augmentation** Since the simulation process of vid2e (Hu et al., 2021) involves temporally upsampling RGB frames to generate dense intermediate frames for event stream generation, it naturally enables the simulation of different levels of motion blur by integrating the dense intermediate

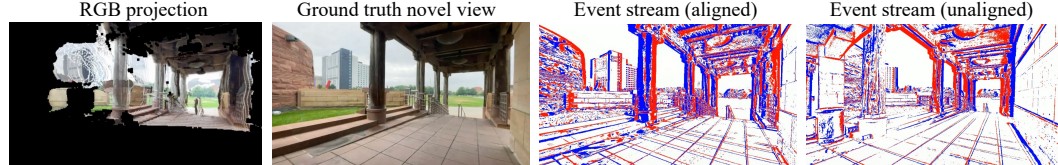

| RGB projection | Ground truth novel view | Event stream (aligned) | Event stream (unaligned) |

Figure 9: **Examples of camera trajectory augmentation.** To encourage the model to synthesize novel views beyond the constraints of the event camera trajectory, we introduce controlled temporal augmentations to the event stream, including forward/backward time shifts and temporal reversal, which result in unaligned event stream for training. While RGB projections provide appearance and coarse viewpoint priors, they suffer from severe occlusions and distortions. In contrast, the event stream, despite being unaligned and appearance-free, offers temporally dense and geometry-rich signals. By leveraging both modalities, our model learns to generate high-quality novel views in an event camera trajectory-agnostic manner.

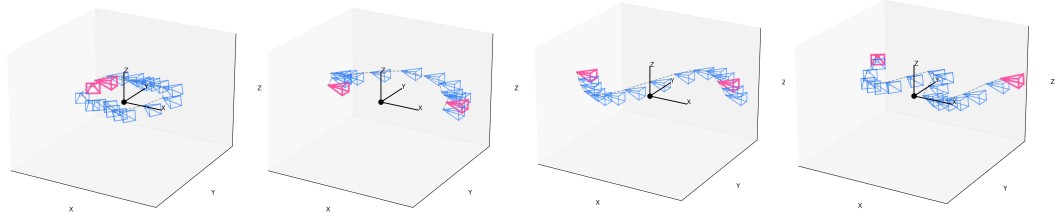

Figure 10: **Examples of evaluation camera trajectories.** We visualize input cameras in red and test cameras in blue; the test cameras are not constrained to the visible regions of the input views and include substantial unseen areas.

frames (Wang et al., 2023; Zhao et al., 2024). We augment our dataset with both sharp and motion-blurred RGB frames, allowing our model to robustly handle motion-blurred inputs.

**Camera Trajectory Augmentation** To encourage our model to synthesize novel views beyond the constraints of the event camera trajectory, we augment the event stream by introducing controlled temporal shifts. Specifically, we extend the event stream forward or backward in time relative to the ground truth novel views, or apply temporal reversal, resulting in deliberate misalignment between the event data and the ground truth novel views. This augmentation forces the model to rely on trajectory-agnostic structural cues from the event stream. Examples of the event stream, RGB projections, and ground-truth novel views are shown in Fig. 9. It can be observed that the RGB projections provide appearance and camera viewpoint information in the target views, but suffer from substantial occlusions and spatial distortions. In contrast, the event stream, despite being unaligned with the ground-truth novel views and lacking color information, offers clear and temporally dense geometric cues of the scene structure. By leveraging both modalities to train the model, our model is able to synthesize high-quality novel views that preserve both appearance fidelity and structural consistency.

## B MORE EXPERIMENTS

### B.1 DETAILED EVALUATION SETTINGS

For the in-the-wild scene comparison, we use 140 test scenes from the DL3DV benchmark (Ling et al., 2024) that do not overlap with our training data, as well as 10 scenes from the Tanks-and-Temples (T&T) (Knapitsch et al., 2017) dataset, with an event simulation contrast threshold of 0.2. For the real event data comparison, since there are no existing novel view synthesis benchmarks that include both sharp RGB frames and event data, we filter out 7 static sequences from the DSEC (Gehrig et al., 2021) driving scene dataset that contains both sharp RGB frames and real-captured event data for evaluation. Given a test novel view sequence from the evaluation dataset, we conduct experiments under 2, 4, and 6 input view settings. We define the view range as the number of frames between two

Table 4: Computation cost comparison with the baselines.

| Method | Upstream MVS | Time (h) | | | Memory (GB) | |
|---|---|---|---|---|---|---|
| | | Infer. | Opt. | Total | Infer. | Opt. |
| E-NeRF (Klenk et al., 2023) | COLMAP (300+ s) | 0.12 | 3.50 | 3.62 | 20 | 26 |
| Event3DGS (Han et al., 2024) | COLMAP (300+ s) | 0.0002 | 0.80 | 0.8002 | **3** | 12 |
| NVS-Solver (You et al., 2025) | COLMAP + DA (300+ s) | 0.18 | 0 | 0.18 | 21 | 0 |
| ViewCrafter (Yu et al., 2025c) | DUSt3R (5 s) | 0.06 | 0 | 0.06 | 24 | 0 |
| Ours | VGGT (1 s) | 0.03 | 0 | **0.03** | 28 | **0** |

Table 5: Ablation on robustness of our method to motion-blurred input.

| Method | PSNR ↑ | SSIM ↑ | LPIPS ↓ |
|---|---|---|---|
| NVS-Solver (You et al., 2025) | 16.25 | 0.539 | 0.407 |
| ViewCrafter (Yu et al., 2025c) | 16.96 | 0.580 | 0.391 |
| Ours w/o event feature | 17.62 | 0.629 | 0.355 |
| Ours | **21.54** | **0.713** | **0.230** |

input views selected from the test sequence. Specifically, the view range for the 2-view setting is 400 frames on DL3DV, 300 frames on Tanks-and-Temples, and 50 frames on DSEC. As shown in Fig. 10, the test views are not limited to visible regions between the input views but also cover substantial unseen regions.

## B.2 OPTIMIZATION AND INFERENCE TIME COMPARISON

We report the per-scene optimization time, inference time, and GPU memory usage of all baselines on a single NVIDIA A100 GPU (40 GB) in Table 4. Although our method employs a diffusion model during inference, it still achieves substantially lower overall runtime compared to optimization-based baselines, owing to its optimization-free design. Moreover, our inference pipeline generates 49 frames in a single forward pass, while each evaluation sequence contains fewer target views than the number of frames produced. As a result, the runtime remains identical across the 2-, 4-, and 6-view input settings.

## B.3 ROBUSTNESS TO OUT-OF-TRAJECTORY EVENT STREAM

In this experiment, we investigate the robustness of EA3D against varying degrees of misalignment between the novel view synthesis trajectory and the event camera trajectory. This analysis is essential for assessing the general applicability of our method to unconstrained real-world scenarios, where the precise camera poses of event data are often unavailable and do not align with the target novel view trajectory. To this end, we perform a controlled ablation by introducing deviations between the novel view trajectory and the event camera trajectory.

Specifically, we construct a series of synthetic event sequences where the event camera poses are perturbed away from the novel view trajectory by varying amounts. The degree of mismatch is quantified using the Absolute Trajectory Error (ATE), computed as the average Euclidean distance between the event camera poses and the corresponding novel view poses. We normalize the ATE to range from [0, 1] for clarity. We evaluate EA3D's novel view synthesis performance under increasing levels of ATE between event camera trajectory and novel view trajectory, while keeping the RGB inputs and target views fixed. As shown in Fig. 11, thanks to our training strategy, the PSNR remains relatively stable even when the event and novel view trajectories are significantly misaligned. These results demonstrate that our EA3D does not rely on precise correspondence between the event camera trajectory and the novel view trajectory. Instead, the model effectively distills trajectory-agnostic geometric cues from the event stream, enabling robust synthesis across diverse and unaligned camera trajectory.

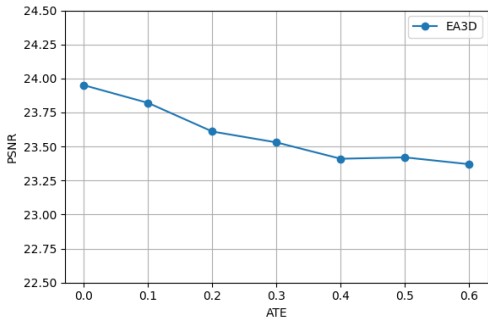 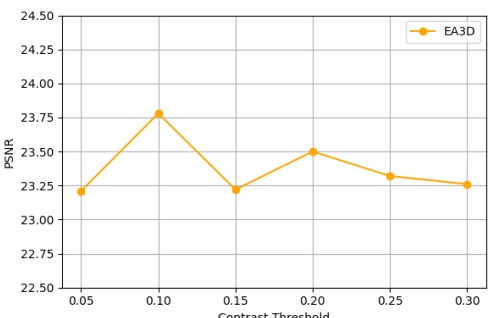

Figure 11: Ablation on robustness to misalignment between novel view trajectory and event camera trajectory.

Figure 12: Ablation on robustness to different Contrast Threshold.

Table 6: Ablation on robustness of our method to fast motion.

| Method | PSNR ↑ | SSIM ↑ | LPIPS ↓ |
|---|---|---|---|
| NVS-Solver (You et al., 2025) | 16.56 | 0.547 | 0.381 |
| ViewCrafter (Yu et al., 2025c) | 16.81 | 0.573 | 0.384 |
| Ours w/o event feature | 17.10 | 0.615 | 0.367 |
| Ours | **22.05** | **0.725** | **0.221** |

## B.4 ROBUSTNESS TO MOTION BLUR

Our Event-DL3DV dataset is simulated with the vid2e event generator (Hu et al., 2021), which temporally upsamples the RGB frames before integrating them to synthesize event streams. The intermediate frame accumulation naturally produces realistic motion blur, and we therefore expose EA3D to both sharp and blurred RGB inputs during training so that the model remains reliable when motion blur is present at test time. To quantify the benefit of event guidance under motion blur, we evaluate on the EvDeblurNeRF-DAVIS dataset (Cannici & Scaramuzza, 2024b), which provides real event streams aligned with motion-blurred RGB images. We compare our full model with RGB-only baselines (NVS-Solver, ViewCrafter, and an EA3D variant without event features) under the 2-view input setting. As shown in Table 5, incorporating events substantially improves reconstruction fidelity in motion blur scenarios.

## B.5 ROBUSTNESS TO FAST MOTION

To further demonstrate the advantage of using event cameras over standard 3D generative models under fast camera motion, we additionally conduct a comparison on 10 drone-captured sequences with rapid motion from the M3ED (Chaney et al., 2023) dataset. As shown in Table 6, incorporating events substantially improves novel view synthesis quality in these fast-motion scenarios.

## B.6 ROBUSTNESS TO CONTRAST THRESHOLD

To evaluate the robustness of EA3D to different Contrast Threshold, we conduct an ablation study. Specifically, we vary the Contrast Threshold from 0.05 to 0.3 using event simulator (Hu et al., 2021). As shown in Fig. 12, EA3D maintains stable PSNR performance across all Contrast Threshold settings, with only minor fluctuations as the threshold varies. We attribute this robustness to our training strategy, which incorporates a diverse set of simulated event streams generated with thresholds sampled from a wide range, thereby enabling the model to generalize across diverse event streams.

## B.7 GENERATIVE ABILITY EVALUATION

Since EA3D leverages event observations that are inaccessible to purely RGB-based methods, we further evaluate perceptual quality metrics that do not require aligned ground-truth, using FID on the

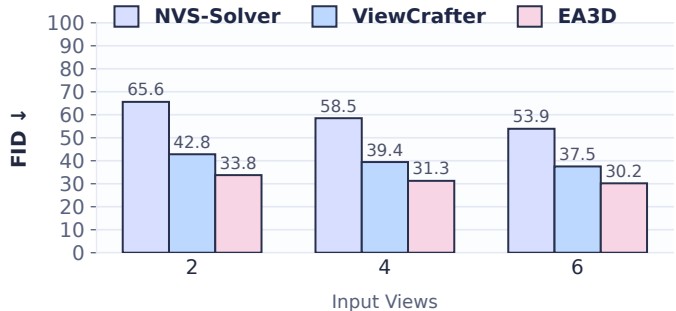

Figure 13: FID comparison (lower is better) on Tanks-and-Temples under varying input views.

Table 7: Comparison with event-based frame interpolation methods.

| Method | PSNR ↑ | SSIM ↑ | LPIPS ↓ |
|---|---|---|---|
| VDM-EVFI (Chen et al., 2024a) | 18.36 | 0.665 | 0.350 |
| Ours | **22.05** | **0.725** | **0.221** |

Tanks-and-Temples dataset. As shown in Fig. 13, we compare EA3D against RGB-based generative NVS methods, NVS-Solver (You et al., 2025) and ViewCrafter (Yu et al., 2025c), under 2-, 4-, and 6-view input settings. Across all settings, EA3D achieves consistently lower FID scores than both baselines, highlighting its superior generative capability.

### B.8 COMPARISON WITH EVENT-BASED FRAME INTERPOLATION METHODS

To further validate the effectiveness of our method in leveraging event data for novel view synthesis under fast camera motion, we compare EA3D with the event-based frame interpolation method VDM-EVFI (Chen et al., 2024a) on 10 drone-captured sequences with fast motion from the M3ED (Chaney et al., 2023) dataset. As shown in Table 7, EA3D consistently achieves higher novel-view synthesis quality than VDM-EVFI.

## C BROADER IMPACT

Event-augmented novel view synthesis has the potential to advance a wide range of downstream applications that require reliable 3D reconstruction in high-speed camera motion or from sparse captured data. The asynchronous and low-latency nature of event streams makes them particularly well-suited for scenarios involving rapid motion or limited imaging bandwidth. As such, our EA3D has the potential to benefit fields such as 3D mapping in autonomous aerial or ground robots, emergency response in visually degraded conditions, and minimally invasive medical imaging where traditional camera systems may face physical or temporal constraints. In addition, the ability to generate coherent 3D scenes from few input views aligns with the growing demand for lightweight sensing in edge devices and wearable systems. It can also support content creation and virtual environment modeling in AR/VR settings, reducing the reliance on dense multi-camera rigs. We encourage responsible use of this technology and recommend that its deployment follow ethical and legal guidelines, especially in surveillance-adjacent or sensitive applications. To support further research, we will release both our codebase and the dataset to the community. In addition, we are developing robust watermarking mechanisms to ensure the traceability and integrity of novel views synthesized by our model.

