# OpenReview forum: "EA3D: Event-Augmented 3D Diffusion for Generalizable Novel View Synthesis"
_ICLR.cc/2026/Conference — ICLR 2026 Poster_

### Official Review · Reviewer_osNE · 2025-10-29

**Soundness:** 3
**Presentation:** 3
**Contribution:** 3
**Rating:** 6
**Confidence:** 3

**Summary:**

The paper proposes EA3D, a generalizable event-augmented 3D diffusion framework for novel view synthesis from sparse RGB frames plus continuous event streams. An EA-Renderer fuses posed RGB appearance features with geometry features extracted from adaptively sliced event voxel grids via cross-attention, producing view-conditioned 3D features that condition a CogVideoX-style 3D-aware diffusion model. The authors also curate Event-DL3DV, pairing synthetic event streams with multi-view RGB and depth. Experiments on DL3DV and Tanks-and-Temples with 2/4/6 views, plus real events from DSEC, report consistent gains over optimization-based event methods (E-NeRF, Event3DGS) and RGB-only generalizable methods (ViewCrafter, NVS-Solver). Ablations indicate benefits from event-derived geometry, adaptive slicing, and a feature reconstruction loss.

**Strengths:**

1.The authors purposed really clear and robust architecture. EA-Renderer elegantly aligns pose-free event geometry with pose-conditioned RGB appearance features in camera trajectory space, directly addressing occlusion and large-baseline challenges. The proposed Event-DL3DV uses randomized thresholds/resolutions to diversify events and adds blur augmentation and temporal reversal to enhance robustness and trajectory generalization.

2.The authors systematically isolate the effects of event geometry, adaptive slicing, and reconstruction loss, showing that event features significantly stabilize NVS under extended baselines.

3.They report inference and memory comparisons with E-NeRF, Event3DGS, NVS-Solver, and ViewCrafter, showing clear runtime advantages.

**Weaknesses:**

1.The paper includes both event-based (E-NeRF, Event3DGS) and non-event diffusion-based (ViewCrafter, NVS-Solver) baselines, which is good coverage. However, it would further strengthen the empirical section to discuss or include newer event-augmented Gaussian frameworks. Considering that many latest works have not yet released code, this omission is understandable and not a fatal limitation, but at least a discussion of such methods would provide better context for positioning EA3D among concurrent developments.

2.While the proposed “geometry feature from event streams” introduces an interesting idea of leveraging voxelized event information to encode geometric continuity, the paper does not elaborate on its structure or behavior in sufficient depth. The feature appears to be embedded within the event encoder of EA-Renderer but lacks standalone analysis, visualization, or theoretical justification. Consequently, although this feature may contribute meaningfully to performance gains (as suggested by Table 3), its novelty and generalizability remain underexplored. A more detailed study would greatly strengthen this contribution.

3.Because the paper does not fully characterize the geometry feature or its underlying mechanism, the overall framework risks being perceived as a system-level composition of existing modules (event voxel encoding, cross-attention fusion, and diffusion-based rendering) rather than a fundamentally new algorithmic contribution. The work would benefit from a clearer articulation of what specific modeling insight or formulation distinguishes EA3D from prior event-guided 3D reconstruction methods.

**Questions:**

The paper introduces several key components, such as the geometry feature extracted from event streams, the adaptive slicing strategy within the event encoder, and the 3D-aware diffusion model. However, these are all described only briefly. Could the authors provide more detailed explanations, figures, or analyses to clarify how these modules operate and how they differ from existing event-based voxel encoders or diffusion frameworks?

---

> ### Author Response · Authors · 2025-11-23
> **Response to Reviewer osNE (1/2)**
>
> Dear Reviewer osNE,
>
> We sincerely appreciate your valuable comments and questions, which have greatly helped us improve our work. We address your concerns as follows.
> ***
> > **Q1:** Discuss newer event-augmented Gaussian frameworks.
>
> **A1:**
> Thank you for this thoughtful suggestion. In the revised manuscript, we have expanded "Section 2 RELATED WORKS-2.1 EVENT CAMERAS (line 097)" to discuss more recent event-augmented 3DGS and NeRF frameworks. While these methods further demonstrate the benefits of event streams for novel view synthesis, they still rely on optimizing a separate scene-specific representation and are not designed to serve as a general, training-free model for event-augmented novel view synthesis across diverse scenes, which is precisely the focus of EA3D.
>
>
>
> ***
> > **Q2:** More detailed description on geometry feature from event streams.
>
> **A2:**  Thank you for the suggestion. We would like to clarify that, in the previous manuscript, we have described the geometry feature from event streams in "Appendix A.1 IMPLEMENTATION DETAILS - Event Stream Processing (line 868)" and "Appendix A.1 IMPLEMENTATION DETAILS - Event Encoder (line 883)", where we describe how the event streams are voxelized and pre-processed, the network architecture of the Event Encoder, the shape of the event features and how the event features are fused with the appearance features. In the revised manuscript, we further visualize the event features in Figure 8 (line 957), which show that the event features effectively extract geometric information from the event stream.
>
> For the generalization ability of event features, in "Section 4.2 COMPARISON RESULTS - Real Event Data Comparison", we evaluate it on real-world driving scenarios; in "Appendix B.4 ROBUSTNESS TO MOTION BLUR", we evaluate it on the EvDeblurNeRF dataset (Cannici & Scaramuzza, 2024b) under real-world scenes with motion blur. In both cases, the event features generalize well and provide significant performance gains. To further demonstrate the generalization ability of the event features, we additionally evaluate on 10 drone-captured sequences with rapid motion from the M3ED dataset (Chaney et al., 2023). As shown in Table 6 (line 1095), incorporating event features consistently improves novel view synthesis quality.
>
> ***
> > **Q3:**  Provide a clearer articulation of what specific modeling insight or formulation distinguishes
> EA3D from prior event-guided 3D reconstruction methods.
>
> **A3:**
> Thank you for the suggestion.  Prior event-guided 3D reconstruction methods inject event streams into scene-specific NeRF/3DGS optimizations: they learn per-scene representations that cannot generalize to new environments. EA3D is instead built as a generalizable model for event-augmented novel view synthesis: we train once on our curated large-scale Event-DL3DV dataset and apply the resulting model to unseen scenes without any per-scene optimization. Three design choices make this possible:
>
> 1. Event-Augmented Feature Renderer (EA-Renderer).
> Rather than only using events as auxiliary losses like in prior works, we condition the model directly on event information. The learnable EA-Renderer constructs view-dependent 3D features in the target camera frustum by fusing posed appearance features with geometry features extracted from unaligned event streams, enabling robust fusion between asynchronous event and RGB signals.
>
> 2. 3D-informed diffusion model.
> Instead of training per-scene NeRF/3DGS models, we start from a video diffusion backbone and condition it on the implicit 3D features produced by the EA-Renderer. A reconstruction loss aligns these implicit 3D features with the VAE features of the ground-truth novel view, tightly coupling geometry learning with the diffusion model and supporting generalization.
>
> 3. Event-DL3DV dataset.
> We curate 10,000 multi-view sequences paired with event streams and depth maps, providing the scale needed to train a single model that generalizes without per-scene optimization.
>
>
> ***
> Unfinished. Please keep reading the comments.

---

> ### Author Response · Authors · 2025-11-23
> **Response to Reviewer osNE (2/2)**
>
> > **Q4:** Provide more detailed explanations/figures for the geometry feature from event streams, adaptive slicing, and the 3D-aware diffusion model.
>
> **A4:**
> Thank you for the suggestion. As decribed in **Q1**, we already provide detailed module descriptions in "Appendix A.1 IMPLEMENTATION DETAILS – Event Stream Processing (line 868)" and "Appendix A.1 IMPLEMENTATION DETAILS – Event Encoder (line 883)", which describes adaptive event slicing strategy, the Event Encoder architecture, feature shapes, and how event features are fused with appearance features. Furthermore, in the revised manuscript, Figure 8 (line 957) visualizes the learned event features, showing that they capture meaningful geometric structure from the event stream.  "Appendix A.1 IMPLEMENTATION DETAILS –Feature Fusion" details how the EA-Renderer align unposed event features with appearance features . Since the video diffusion backbone follows the standard CogVideoX architecture, so we do not redraw it and instead focus on how our conditioning connects to it.

---

### Official Review · Reviewer_BEQ9 · 2025-10-30

**Soundness:** 3
**Presentation:** 3
**Contribution:** 2
**Rating:** 4
**Confidence:** 3

**Summary:**

The paper introduces EA3D, a framework for generalizable novel view synthesis that uses sparse RGB frames and continuous event streams as input. It proposes a Event-Augmented Feature Renderer module, which is a learnable module that constructs 3D features for the target camera path. It extracts appearance cues from the sparse RGB frames and captures geometric structure from estimated camera parameters and depths. The 3D features from the EA-Renderer are used to condition a video diffusion model (based on CogVideoX) for novel view synthesis.

**Strengths:**

1. Extensive experiments: The paper provides comprehensive comparisons with baselines, investigates the impact of varying the number of input samples, and includes relevant ablation studies on model design.
2. The creation of the large-scale Event-DL3DV dataset would be a valuable contribution if it is made open source.
3. The proposed method achieves noticeable improvements in both visual quality and quantitative metrics.

**Weaknesses:**

1. Reliance on Upstream Models: The EA-Renderer requires camera parameters and depth maps for the input RGB frames, which are obtained from an "off-the-shelf multi-view stereo model". This means EA3D is not fully end-to-end, and its performance is dependent on the accuracy of this prerequisite model.
2. The idea of incorporating explicit 3D information into diffusion models has been explored in many previous works, and this paper does not present particularly novel aspects in this regard.

**Questions:**

In Table 4, "Computation cost comparison with the baselines," does the reported computation cost include the time required to estimate camera parameters and depths using an off-the-shelf multi-view stereo model?

---

> ### Author Response · Authors · 2025-11-23
> **Response to Reviewer BEQ9**
>
> Dear Reviewer BEQ9,
>
> We would like to express our sincere gratitude for your valuable and insightful comments and questions, which have greatly contributed to improving our work. We address your concerns as follows.
> ***
> > **Q1:** The creation of the large-scale Event-DL3DV dataset would be a valuable contribution if it
> is made open source.
>
> **A1:**
> Thank you for the suggestion, we will make Event-DL3DV dataset and our code and model publicly available upon acceptance of the paper.
>
>
> ***
> > **Q2:** Reliance on Upstream MVS Models.
>
> **A2:**  Thank you for the comments. We would like to clarify that relying on an off-the-shelf multi-view stereo (MVS) model for camera poses and depth maps/point clouds is standard practice in almost all novel view synthesis methods. In our comparison baselines, ENeRF and Event3DGS use the non-learning-based COLMAP for poses and point clouds, NVS-Solver uses COLMAP + DepthAnything for poses and depths, ViewCrafter uses the learning-based DUSt3R, and our method uses the learning-based VGGT.
> In practice, COLMAP is often fragile under sparse views and motion-blurred input. As a result, traditional sparse-view NeRF/3DGS pipelines first run COLMAP on dense multi-view sequences to recover reliable poses and point clouds, and then train using only a sparse subset of those views to match the sparse-view novel view synthesis setting; our Event3DGS and ENeRF baselines follow exactly this procedure in our experiments. However, this experimental setting requires access to dense views, which is impossible in real-world fast motion scenes, where only sparse views are available.
> Consequently, recent works like ViewCrafter advocate using learning-based MVS models such as DUSt3R for more robust point cloud and camera pose initialization. Our method further uses VGGT as an upstream MVS model, which is in fact more robust than COLMAP in pose and point cloud estimation.
>
>
> ***
> > **Q3:**  The idea of incorporating explicit 3D information into diffusion models has been explored in many previous works, and this paper does not present particularly novel aspects in this regard.
>
> **A3:** Thank you for the comments. We would like to clarify that while prior works have explored incorporating explicit 3D information into diffusion models, our work specifically studies the limitations of such RGB-only 3D-guided approaches (e.g., ViewCrafter and NVS-Solver, which we adopt as our main baselines) under large-baseline novel view synthesis and fast camera motion settings. In these challenging regimes, we show that purely RGB-based explicit 3D guidance often suffers from severe artifacts and instability.
>
> In contrast, one of the core contributions of our work is the design of EA-Renderer, which unifies temporally continuous event information and RGB information into an implicit 3D feature representation. Compared to the explicit 3D information used in previous works, our event-augmented implicit 3D feature provides more occlusion-resilient information with much higher temporal resolution, enabling the model to better handle large-baseline and fast-motion inputs.
> This aspect of contribution is also acknowledged by Reviewer RKQv ("The core idea of leveraging event cameras to provide complementary information for novel view synthesis is both novel and compelling") and Reviewer osNE ("EA-Renderer elegantly aligns pose-free event geometry with pose-conditioned RGB appearance features in camera trajectory space, directly addressing occlusion and large-baseline challenges.").
>
> Furthermore, through systematic comparisons in "Section 4 EXPERIMENTS", "Appendix B.4 ROBUSTNESS TO MOTION BLUR", and "Appendix B.5 ROBUSTNESS TO FAST MOTION", we demonstrate that incorporating event information yields significant improvements over explicit-3D diffusion baselines (NVS-Solver and ViewCrafter) in large-baseline, fast camera motion, and motion blur settings, demonstrating the effectiveness of our proposed method.
>
>
> ***
> > **Q4:** Computation time comparison with the baselines.
>
>  **A4:**
> Thank you for the question. In the previous version, the computation time reported in Table 4 for all methods did not include the runtime of the upstream MVS models. In the revised Table 4, we add additional columns to report the cost of these upstream MVS models. As shown in the updated Table 4 (line 1027), the upstream model used by our method is in fact the fastest among all compared approaches.

---

> ### Comment · Reviewer_BEQ9 · 2025-11-28
>
> I appreciate the authors' efforts in addressing my concerns, particularly regarding the Upstream Models and technical contribution. In light of the rebuttal and the authors' responses to other reviewers, I am happy to raise my final score to 6.

---

### Official Review · Reviewer_RKQv · 2025-10-31

**Soundness:** 2
**Presentation:** 3
**Contribution:** 2
**Rating:** 6
**Confidence:** 3

**Summary:**

The paper proposes an event-augmented framework for novel view synthesis. The core idea is to leverage event cameras to provide complementary information to strengthen the overall synthesis process, especially in scenarios with rapid camera movements. The framework consists of two main components: an event-augmented (EA) feature renderer and a feature-conditioned video diffusion model. The EA feature renderer fuses event information with RGB features using cross-attention in the 3D space, creating a strong feature prior for subsequence generation process. The diffusion model, adapted from CogVideoX, is then conditioned on these features to synthesize novel views in video sequence. Experiments demonstrate the advantages of adopting event features to boost the performance of novel view synthesis.

**Strengths:**

* The core idea of leveraging event cameras to provide complementary information for novel view synthesis is both novel and compelling.
* The proposed framework is straightforward and effective. The experiments clearly demonstrate the benefits of incorporating event data.
* The paper is well-written, clearly structured, and easy to understand

**Weaknesses:**

* A central claim of the paper is that event data is particularly helpful for fast camera movements. However, the experimental section seems to lack a direct evaluation to support this specific claim. It would significantly strengthen the paper to include an analysis comparing performance under fast camera motion.
* As event data has strong correlation with camera trajectories, I am curious about the model's generalization capabilities under different camera trajectories. How does the method perform when tested with camera trajectories that are different from those seen during training (e.g., different paths or motion patterns)? A discussion on this aspect would be very insightful.

**Questions:**

N.A.

---

> ### Author Response · Authors · 2025-11-23
> **Response to Reviewer RKQv**
>
> Dear Reviewer RKQv,
>
> Thank you very much for your insightful comments and questions, which have been extremely helpful for improving our work. We address your concerns as follows.
> ***
> > **Q1:** It would significantly strengthen the paper to include an analysis
> comparing performance under fast camera motion.
>
> **A1:**
> Thanks for your suggestion.
> We would like to clarify that we have included an experiment on scenes with fast motion and motion blur in "Appendix B.4 ROBUSTNESS TO MOTION BLUR" using the EvDeblurNeRF-DAVIS dataset (Cannici & Scaramuzza, 2024b). As shown in Table 5 (line 1038) in the appendix, our method achieves substantially improved novel view synthesis quality in these scenarios.
>
> In the revised manuscript, we additionally conduct a comparison on 10 drone-captured fast-motion sequences from the M3ED dataset (Chaney et al., 2023). The results shown below further validate that our method achieves better performance in fast camera motion.
>
> | Method                 | PSNR ↑  | SSIM ↑  | LPIPS ↓  |
> |------------------------|---------|---------|----------|
> | NVS-Solver             | 16.56   | 0.547   | 0.381    |
> | ViewCrafter            | 16.81   | 0.573   | 0.384    |
> | Ours w/o event feature | 17.10   | 0.615   | 0.367    |
> | **Ours**               | **22.05** | **0.725** | **0.221** |
>
>
>
> ***
> > **Q2:** Robustness under varying camera trajectories.
>
> **A2:**  Thank you for the insightful question.
> Firstly, as described in "Appendix A.2 DATASET", our constructed training dataset (Event-DL3DV dataset) consists of 10,000 sequences, containing a large number of diverse camera trajectories. These trajectories differ in paths, speeds, and motion patterns. As a result, our trained model can generalize to arbitrary camera trajectories at test time. The test camera trajectories shown in our demo videos are all unseen during training, yet the model still produces stable and consistent novel views.
>
> Secondly, as discussed in "Appendix A.2 DATASET-Camera Trajectory Augmentation" and Figure 9 (line 978), we explicitly consider the strong coupling between event streams and camera trajectories. During training, we apply trajectory augmentation on the event streams so that the event camera trajectories do not strictly align with the target novel view trajectories. This encourages the model to rely on the underlying 3D geometry encoded in the event features rather than overfitting to a specific trajectory pattern, and it improves robustness to arbitrary test-time trajectories.
>
> Finally, in "Appendix B.3 ROBUSTNESS TO OUT-OF-TRAJECTORY EVENT STREAM", we provide a dedicated experiment where the event-camera trajectory and the novel view trajectory are intentionally misaligned. The results show that our method remains robust under such trajectory mismatch and still clearly outperforms the baselines. These experiments indicate that our model generalizes well to camera trajectories that differ from those seen during training.

---

### Official Review · Reviewer_eG5r · 2025-11-01

**Soundness:** 3
**Presentation:** 4
**Contribution:** 3
**Rating:** 4
**Confidence:** 3

**Summary:**

The paper proposes a generalizable novel view synthesis framework that fuses event streams and RGB images to achieve photorealistic rendering under fast camera motion. The method extracts appearance features from RGB frames using a multi-view stereo model, encodes geometry-aware event features through voxel-based event representation, and integrates them via attention module. Experimental results demonstrate that the proposed fusion module effectively achieves its intended goal and enables the framework to reach state-of-the-art performance in novel view synthesis.

**Strengths:**

1. This paper is the first to propose a generalizable novel view synthesis framework based on event camera,. The overall network design is clear, simple, and effective, leading to strong performance.

2. The use of 3D features to condition the DiT appears both effective and novel, offering a promising direction for improving 3D-aware generation.

3. The final results are quite impressive in term of extremely sparse input conditions, the proposed method even outperforms per-scene optimization approaches.

4. Adaptive Slicing is an effectively strategy for encoding event feature.

5.The paper is well written, clearly structured, and easy to follow

**Weaknesses:**

1. The ablation study appears unfair to me, as the authors leverage VGGT + video diffusion, which could be the primary factor contributing to the performance improvement. From my perspective, as shown in works like FlowR [1], combining geometry foundation model + multi-view diffusion can already yield strong performance. However, the ablation study in this paper does not clearly demonstrate that the event information is fully utilized or that it significantly contributes to the performance gains.

2. From Table 1, the performance improvement becomes less significant when the number of input views increases (e.g., to six views), indicating that the method’s advantage is mainly evident in extremely sparse-view settings.

[1] Fischer, T., Bulò, S. R., Yang, Y. H., Keetha, N., Porzi, L., Müller, N., ... & Kontschieder, P. (2025). Flowr: Flowing from sparse to dense 3d reconstructions. In Proceedings of the IEEE/CVF International Conference on Computer Vision (pp. 27702-27712).

**Questions:**

1. How did you align the VGGT poses to the coordinate system used in your evaluation? I find this part a bit confusing, and I have some concerns about the fairness of the evaluation. If the authors could clarify this alignment process, I would consider increasing my score.

2. Would you mind add one more ablation about without event features? Which will help us to understand why we need to do generalizable event-based novel view synthesis.

3.Is your evaluation performed only between the input views, or over the entire scene including unseen regions? The results are surprisingly strong, so I’m concerned that the evaluation might be limited to the input-view range rather than the full scene.

---

> ### Author Response · Authors · 2025-11-23
> **Response to Reviewer eG5r (1/2)**
>
> Dear Reviewer eG5r,
>
> We sincerely appreciate your valuable comments and questions, which have greatly helped us improve our work. Our responses are as follows.
> ***
> > **Q1:** The ablation study in this paper does not clearly demonstrate that the event information is fully utilized or that it significantly contributes to the performance gains.
>
> **A1:**
> Thank you for the question. We would like to clarify that, in our previous manuscript, we have already conducted the following ablations to demonstrate that the event information makes a significant contribution to the performance.
>
> 1. In "4.3 ABLATION STUDY – Effectiveness of Geometry Features from Event Streams", we perform a controlled comparison between our full model and ours w/o geometry feature (i.e., without event information). The quantitative results in Table 3 (line 438) show that injecting geometry feature (event information) leads to a 4.6 dB PSNR gain, and the qualitative results in Figure 4 (w/o Geometry Feature) further validate that incorporating event information significantly reduces artifacts.
>
> 2. In "Appendix B.4 ROBUSTNESS TO MOTION BLUR", we include an ablation on real fast-motion scenes with motion blur using the EvDeblurNeRF-DAVIS dataset (Cannici & Scaramuzza, 2024b). In this ablation, we compare our full model with ours w/o geometry feature (i.e., without event information), as well as two RGB-only baselines (NVS-Solver and ViewCrafter). As shown in Table 5 (line 1039), incorporating event information substantially improves novel view synthesis quality in these motion-blur scenes, demonstrating that event cameras provide clear benefits over existing RGB-only generative models when fast camera motion and motion blur are present.
>
> As also acknowledged by Reviewer gMy4 (“The ablation studies in Section 4.3 clearly validate the core contribution, and in particular, the main ablation shows that incorporating event-based geometry features significantly improves results”) and Reviewer RKQv ("The experiments clearly demonstrate the benefits of incorporating event data"), we believe that the ablations described above provide solid evidence for the effectiveness of incorporating event information.
>
> To further demonstrate the advantage of using event information, we additionally conduct an ablation on 10 drone-captured fast-motion sequences from the M3ED dataset (Chaney et al., 2023). The results shown below further confirm that event information provides clear benefits in challenging fast-motion scenarios.
>
> | Method                 | PSNR ↑  | SSIM ↑  | LPIPS ↓  |
> |------------------------|---------|---------|----------|
> | Ours w/o event feature | 17.10   | 0.615   | 0.367    |
> | **Ours**               | **22.05** | **0.725** | **0.221** |
>
>
> ***
> > **Q2:**  From Table 1, the performance improvement becomes less significant when the number of input views increases.
>
> **A2:**  Thanks for the comments.
> We quantify the performance gains (PSNR) of our method at 2/4/6 views over Table 1 in the table below.
>
> | Dataset | Method      | 2-view PSNR ↑        | 4-view PSNR ↑        | 6-view PSNR ↑        |
> |---------|-------------|------------------|------------------|------------------|
> | DL3DV   | E-NeRF      | 18.01 (-4.81)    | 22.97 (-1.83)    | 25.19 (-0.22)    |
> |         | Event3DGS   | 16.84 (-5.98)    | 22.10 (-2.70)    | 25.26 (-0.15)    |
> |         | ViewCrafter | 19.10 (-3.72)    | 20.78 (-4.02)    | 22.51 (-2.90)    |
> |         | NVS-Solver  | 17.75 (-5.07)    | 21.83 (-2.97)    | 22.18 (-3.23)    |
> |    | Ours        | 22.82        | 24.80        | 25.41       |
> | T&T     | E-NeRF      | 22.96 (-0.54)    | 25.46 (+0.69)    | 26.21 (+0.37)    |
> |         | Event3DGS   | 22.42 (-1.08)    | 25.54 (+0.77)    | 26.32 (+0.48)    |
> |         | ViewCrafter | 18.24 (-5.26)    | 22.26 (-2.51)    | 22.87 (-2.97)    |
> |         | NVS-Solver  | 17.68 (-5.82)    | 20.57 (-4.20)    | 20.85 (-4.99)    |
> |      | Ours        | 23.50        | 24.77        | 25.84       |
>
> For generative baselines (NVS-Solver and ViewCrafter) that are unable to incorporate event data, the improvements remain significant, even at 4 and 6 views. However, for the optimization-based baselines E-NeRF and Event3DGS, we acknowledge that the performance gains do become less significant.
> The reason is, as described in "4.1 EXPERIMENTAL SETTING-Evaluation Setting (line 350)", in the comparison in Table 1, we use event streams simulated from ground truth novel views as training input of E-NeRF and Event3DGS, since they are designed to synthesize novel views along the event camera trajectory.
> This setup allows E-NeRF and Event3DGS to access aligned and detailed ground truth information from the event stream.
>
> ***
> Unfinished. Please keep reading the comments.

---

> ### Author Response · Authors · 2025-11-23
> **Response to Reviewer eG5r (2/2)**
>
> In contrast, our method is designed to support novel view synthesis along flexible rendering trajectories without requiring strict alignment with the event camera trajectory. As noted in line 355, to validate this capability, we simulate event streams that are misaligned with the ground truth novel views for our model. This creates a more general and challenging setting for our method, as it does not have access to aligned ground-truth information from the event stream, while E-NeRF and Event3DGS can use it.
> Even under this more challenging setting, our generalizable model achieves comparable or better results than the optimization-based E-NeRF and Event3DGS. This is because our model is more effective at handling occlusions and recovering missing regions that are not observed in the input RGB views.
> As the number of training views increases, occlusions and missing regions are reduced, and E-NeRF and Event3DGS can additionally exploit ground-truth event information. Consequently, their performance naturally improves, and the relative gains of our method become smaller.
>
> ***
> > **Q3:**  About how to align the VGGT poses to the coordinate system used in our evaluation.
>
> **A3:**
> Thanks for the question. We would like to clarify that both our method and all baselines are evaluated in their own pose coordinate systems, so no explicit cross-system alignment is required.
>
> Concretely, EventNeRF, E-NeRF, and NVS-Solver all rely on COLMAP for camera pose estimation, so both their input training views and test views are represented in the COLMAP coordinate system. ViewCrafter relies on DUSt3R for pose estimation, so its training and test camera poses are defined in the DUSt3R coordinate system. Similarly, our method relies on VGGT for pose estimation, and thus both the input views and the test views of our method are represented in the VGGT coordinate system.
>
> For each method, we always render novel views and fetch the corresponding ground-truth views using the same camera poses within that method’s own coordinate system. Therefore, what is evaluated is the image reconstruction quality under each method’s internal pose system, and no additional global alignment or manual pose refinement is performed. This ensures that the comparison is fair: every method uses its standard pose estimation pipeline and is evaluated in the coordinate system it naturally operates in.
>
> ***
> > **Q4:**  Add one more ablation about without event features.
>
> **A4:** Thanks for the suggestions. We have provided explanation and more ablation about our method without geometry feature (event feature) in **Q1**.
>
>
> ***
> > **Q5:** Is the evaluation performed only between the input views, or over the entire scene including unseen regions?
>
> **A5:**
> Thanks for the question. Our evaluation is not restricted to visible viewpoints between the input views, it also covers unseen regions.
> As described in "Appendix B.1 DETAILED EVALUATION SETTINGS", our evaluation follows the standard sparse-view novel view synthesis evaluation setting. Specifically, for each test sequence, we sample sparse training/input frames and use intermediate frames as test frames. The sampling interval between input views is large (400 frames on DL3DV, 300 frames on Tanks-and-Temples, and 50 frames on DSEC), so the camera trajectory between the input views is not a simple interpolation, but contains diverse trajectories and includes substantial unseen areas. We provide several visualizations of the train/test camera poses in "Appendix B.1 DETAILED EVALUATION SETTINGS-Figure 10 (line 995)" to illustrate that the test views include substantial unseen areas and not limited to interpolations between the input views.
> This is also part of the reason why our training-free method can achieve comparable or even better performance than optimization-based methods: thanks to the generative capability of our method, it can effectively handle occlusions and missing content in regions that are invisible in the input views.

---

> > ### Comment · Reviewer_eG5r · 2025-11-28
> >
> > Since authors follow the common evaluation setting, my concerns about the evaluation have been mostly addressed, and I would like to raise my score to 6. However, I still have one remaining question. If your coordinate system is established based on VGGT, how do you obtain the poses for the test views? If VGGT is run on the test views to estimate their poses, is there any possibility that information from the test views leaks into the reconstruction process?

---

> ### Author Response · Authors · 2025-11-28
> **Response to Reviewer eG5r**
>
> We appreciate your constructive feedback. For the remaining concern about pose estimation, we would like to clarify that:
>
>  1. Similar to the standard protocol in our comparison baselines E-NeRF and Event3DGS, where COLMAP estimates poses for both training and test views jointly to establish a unified coordinate system, we use VGGT to estimate poses for both input and test views. This ensures that the test poses are aligned within the same coordinate system as the input poses.
>
> 2. To prevent information leakage, we follow the evaluation protocol of our comparison baseline ViewCrafter. We do not simply extract the input view point clouds from the global point cloud (input views + test views) generated during the joint pose estimation process, as such input view point clouds would have enhanced accuracy since VGGT has observed more views. Instead, after the poses are estimated, we independently generate the input point cloud using only the input views. This setup ensures that novel view synthesis relies exclusively on information available from the input views, guaranteeing that no content from the test views leaks into the generation process.

---

### Official Review · Reviewer_gMy4 · 2025-11-03

**Soundness:** 3
**Presentation:** 3
**Contribution:** 2
**Rating:** 8
**Confidence:** 4

**Summary:**

This paper proposes a framework for generalizable novel view synthesis that uses a set of sparse RGB frames and a continuous event stream as input  to generate novel RGB views. The method conditions a video generative model (a Diffusion Transformer based on CogVideoX)  on 3D features. These 3D features are constructed by an "EA-Renderer," which uses an off-the-shelf 3D reconstruction method (VGGT) to get posed appearance features, and then enriches these features with geometric information from event-based voxels. Since poses for the event camera are not available , the method learns a feature fusion module (using cross-attention)  to integrate the un-posed event features into the posed 3D features. The paper provides comparisons on real-world and synthetic datasets against both RGB-only 3D generative models and optimization-based, event-camera-specific methods. The ablation study provides a much-needed demonstration of performance degradation when the event-based stream is removed.

Despite the technical contribution being somewhat incremental (combining existing MVS, video diffusion, and attention mechanisms), the application to event-augmented novel view synthesis is valuable and I am arguing for acceptance. However, I would strongly encourage the authors to address the weaknesses below -- the paper would be far more convincing if the authors added experiments on challenging scenes with real fast motion (e.g. fast-moving objects or drone footage) to truly demonstrate the advantage of using event cameras over standard generative 3D models.

**Strengths:**

- Important, interesting, and timely problem: The paper addresses the challenging problem of novel view synthesis from sparse inputs under fast camera motion -- a scenario where event cameras offer a clear theoretical advantage over standard RGB-only methods and where existing 3D generative model research hasn't focused on .
- Fusion Strategy, though obvious choice and computationally intensive, is an interesting technical contribution. The proposed architecture for fusing posed RGB features with un-posed event features via a cross-attention mechanism  is a logical way to combine these complementary data sources.
- Effective Ablations: The ablation studies in Section 4.3 clearly validate the core contribution, and in particular, the main ablation shows that incorporating event-based geometry features significantly improves results, especially as the baseline (view range) between input frames increases.

**Weaknesses:**

- Dataset Limitations: My main concern is that the evaluation datasets do not fully demonstrate the scenarios where event cameras are organically needed. The paper uses "static sequences" from the real-world DSEC dataset , while other benchmarks (DL3DV, T&T) rely on simulated events. Most motion blur is artificially induced. The paper would be significantly stronger if it outperformed existing 3D generative models in datasets where event cameras are truly necessary (e.g. fast drone videos, rapid human motion).
- Missing Comparisons: The paper does not compare against EGVD (Zhang et al., 2025) or (Chen et al., 2024a). While the authors argue these methods are for interpolation along the trajectory, the evaluations in this paper also appear to synthesize views near the original camera path. A comparison seems feasible and would benefit to fully evaluate the approach.
- Minor: The method is heavily reliant on an off-the-shelf pretrained multi-view 3D reconstruction method (VGGT)  to extract the initial 3D point cloud and posed features.
- Minor: Inaccuracies in Related Work (Sec 2.2): The related work section needs revision for accuracy. 1) The claim that ReconFusion / ZeroNVS "struggles with pose control... due to implicit pose encoding"  is incorrect as there are explicit 3D method. 2) The use of "3D-aware diffusion model"  is confusing. This term typically refers to models that are 3D-consistent by design (e.g. by incorporating a renderer in the denoising loop, like EG3D DMV3D and RenderDiffusion, which haven't been cited), which is not the case here. Therefore, the claim "To ensure 3D consistency"  should be weakened to "To encourage 3D consistency," as a video diffusion model does not guarantee this. 3) The claim that other video diffusion models "degrade when the input views exhibit large viewpoint variations" is a weak argument; methods like CAT3D perform well from very sparse views. The primary argument against them should be their inability to leverage event data.

**Questions:**

- The feature fusion module performs cross-attention between each appearance feature and the entire event feature set. How does the computational cost of this fusion scale, and does it limit the practical length of the event streams you can process?
- How robust is the method to failures in the initial MVS (VGGT) step, especially in texture-less regions or under extreme motion?

---

> ### Author Response · Authors · 2025-11-23
> **Response to Reviewer gMy4 (1/2)**
>
> Dear Reviewer gMy4,
>
> Many thanks for your valuable comments and questions, which help us a lot to improve our work. We address your questions as follows.
> ***
> > **Q1:** The paper would be significantly stronger if it outperformed existing 3D generative models in datasets with real fast motion
>
> **A1:**
> Thanks for your suggestion.
> We would like to clarify that we have included an experiment on scenes with fast motion and motion blur in "Appendix B.4 ROBUSTNESS TO MOTION BLUR", using the EvDeblurNeRF-DAVIS dataset (Cannici & Scaramuzza, 2024b). In this setting, we compare our full EA3D model with generative baselines (NVS-Solver, ViewCrafter, and an EA3D variant without event features). As shown in Table 5 (line 1039), our method achieves significantly improved novel view synthesis quality in these scenarios.
> To further demonstrate the advantage of using event cameras over standard 3D generative models under fast camera motion, we additionally conduct a comparison on 10 drone-captured fast-motion sequences from the M3ED dataset (Cannici & Scaramuzza, 2024b). The results below further validate the advantage of our method.
>
> | Method                 | PSNR ↑  | SSIM ↑  | LPIPS ↓  |
> |------------------------|---------|---------|----------|
> | NVS-Solver             | 16.56   | 0.547   | 0.381    |
> | ViewCrafter            | 16.81   | 0.573   | 0.384    |
> | Ours w/o event feature | 17.10   | 0.615   | 0.367    |
> | **Ours**               | **22.05** | **0.725** | **0.221** |
>
>
>
> ***
> > **Q2:**  Add comparison with event-based frame interpolation methods.
>
> **A2:**  Thanks for the suggestion.
> To further validate the effectiveness of our method in leveraging event data for novel view synthesis
> under fast camera motion, we compare EA3D with the event-based frame interpolation method VDM-EVFI (Chen et al., 2024a) on 10 drone-captured sequences with fast motion from the M3ED (Chaney
> et al., 2023) dataset. As shown in the table below, EA3D consistently achieves higher novel-view synthesis
> quality than VDM-EVFI.
>
>
> | Method        | PSNR ↑ | SSIM ↑ | LPIPS ↓ |
> |--------------|--------|--------|---------|
> | VDM-EVFI     | 18.36  | 0.665  | 0.350   |
> | **Ours**     | **22.05** | **0.725** | **0.221** |
>
>
> Importantly, VDM-EVFI is designed for interpolation along the event streams, assuming events are captured on the target camera trajectory, whereas EA3D can handle out-of-trajectory novel views. As shown in "Appendix B.3 ROBUSTNESS TO OUT-OF-TRAJECTORY EVENT STREAM", EA3D remains effective when the target camera trajectory deviates from the event camera trajectory, while interpolation-based methods like VDM-EVFI cannot be directly applied.
>
>
> ***
> > **Q3:**  Reliance on an off-the-shelf pretrained multi-view 3D
> reconstruction method (VGGT) to extract the initial 3D point cloud and posed features.
>
> **A3:**
> Thanks for the comment. We would like to clarify that relying on an off-the-shelf multi-view stereo (MVS) model to extract the initial 3D point cloud and camera poses is standard practice in novel view synthesis methods: NeRF, 3DGS and their event-based variants (ENeRF, Event3DGS) use the classical, non-learning-based COLMAP for poses and point clouds, NVS-Solver uses COLMAP + DepthAnything, ViewCrafter uses the learning-based DUSt3R, and our method uses the learning-based VGGT.
> In practice, COLMAP is often fragile under sparse views and motion-blurred input. As a result, traditional sparse-view NeRF/3DGS pipelines first run COLMAP on dense multi-view sequences to recover reliable poses and point clouds, and then train using only a sparse subset of those views to match the sparse-view novel view synthesis setting; our Event3DGS and ENeRF baselines follow exactly this procedure in our experiments.
> However, this experimental setting requires access to dense views, which is inaccessible in real fast-motion scenarios, where in practice only sparse views are available.
> Consequently, recent works like ViewCrafter advocate using learning-based MVS models such as DUSt3R for more robust point cloud and camera pose initialization. Our method follows the same practice by using VGGT as an off-the-shelf learning-based MVS model, without introducing any additional assumptions beyond those commonly adopted in prior work.
>
> ***
> > **Q4:**  Correct related work descriptions.
>
> **A4:** Thanks for pointing out these issues. We revised the related work section to (1) correct the descriptions of ReconFusion and ZeroNVS, (2) replace "3D-aware diffusion model" with "3D-informed diffusion model" and soften "ensure 3D consistency" to "encourage 3D consistency" and (3) remove the claim that other video diffusion models "degrade" under large viewpoint changes, instead emphasizing that their main limitation in our context is the inability to leverage event data to handle fast camera motion and motion blur.
>
> ***
> Unfinished. Please keep reading the comments.

---

> ### Author Response · Authors · 2025-11-23
> **Response to Reviewer gMy4 (2/2)**
>
> > **Q5:** Computational cost of Event-Appearance feature cross-attention.
>
>  **A5:**
> Thanks for raising this question. We compare the computational cost of our introduced cross-attention operation with the native 3D self-attention inherited from the base model CogVideoX to demonstrate its efficiency.
>
> As detailed in "Appendix A.1 MODEL ARCHITECTURE", we first apply an adaptive event slicing strategy to preprocess the dense event stream and convert it into $N$ slices, which already reduces the computational cost substantially compared to operating on raw events. These slices are then encoded and compressed into an event feature volume
> $F_{\text{event}} \in \mathbb{R}^{\frac{N}{8}\times\frac{H}{8}\times\frac{W}{8}\times C}$,
> which is reshaped into
> $N_2 = \frac{N}{8}\cdot\frac{H}{8}\cdot\frac{W}{8}$ key/value tokens for the cross-attention module.
> The appearance feature has size $F_{\text{appr}}^{i} \in \mathbb{R}^{\frac{H}{8}\times\frac{W}{8}\times C}$ and is flattened into
> $N_1 = \frac{H}{8}\cdot\frac{W}{8}$ query tokens. Therefore, the cross-attention operation has complexity
> $
> \mathcal{O}(N_1 N_2)
> = \mathcal{O}\left(\frac{N}{8}\left(\frac{H}{8}\cdot\frac{W}{8}\right)^2\right).
> $
> For a fixed spatial resolution $H \times W$, the complexity thus scales linearly with $\frac{N}{8}$. During inference, this cross-attention is applied only once to obtain the fused 3D feature, which is then injected into the video diffusion model.
>
> In comparison, the 3D self-attention in the Diffusion Transformer operates on the whole video sequence with token count $\frac{N}{4}\cdot\frac{H}{8}\cdot\frac{W}{8}$, so its complexity is
> $
> \mathcal{O}\left(\left(\frac{N}{4}\cdot\frac{H}{8}\cdot\frac{W}{8}\right)^2\right),
> $
> which is quadratic in $\frac{N}{4}$. Moreover, this 3D self-attention is executed repeatedly in every Diffusion Transformer block and at every diffusion step during inference, leading to hundreds of such operations per sample.
>
> Therefore, compared to the 3D self-attention layers inherited from the base model CogVideoX, our introduced cross-attention operation accounts for only a small fraction of the overall computational cost and does not become a practical bottleneck.
>
> ***
> > **Q6:** Robustness of the method to failures in the initial MVS (VGGT) step.
>
>  **A6:**
> Thanks for the question. As discussed in **Q3**, relying on an off-the-shelf multi-view stereo (MVS) model for initial poses and geometry is standard practice in novel view synthesis; classical NeRF/3DGS-based methods typically use COLMAP, while our method uses the learning-based VGGT. In practice, VGGT is empirically more robust than COLMAP under sparse views and in low-texture regions.
> Nevertheless, we agree that under extreme conditions, such as extremely textureless scenes, VGGT can also fail, and in such cases all comparison baselines and our method may degrade. We acknowledge this dependence on the initial MVS step as a limitation of our current approach and have added this point to the revised limitations section.

---

### Author Response · Authors · 2025-11-24
**We have updated the revised PDF and responsed to every reviewer**

Dear Reviewers,

We deeply appreciate the time and effort you have dedicated to providing detailed and insightful feedback on our work. Your comments have been invaluable in improving the quality of our paper. Detailed responses to each point are provided under the corresponding reviewer’s comments.

---

### Author Response · Authors · 2025-11-29
**Summary of Responses**

Dear Area Chair,

We sincerely thank you for handling our submission and the reviewers for their constructive feedback. We have responded to all reviewers, conducting additional experiments and revising the manuscript to address their concerns. Below is a summary of our responses and discussion.

---
- Reviewer gMy4 (Initial Score: 8) highlighted that our work addresses an "important, interesting, and timely problem" with "effective ablations". To address their request for evaluation on real fast-motion scenes, we pointing to our existing experiments on the EvDeblurNeRF-DAVIS dataset and added new experiments on drone-captured sequences from the M3ED dataset, where our method significantly outperformed baselines. We also included the requested comparison with the event-based interpolation method VDM-EVFI, demonstrating superior synthesis quality, and clarified the computational cost of our cross-attention module.

---
- Reviewer eG5r (Initial Score: 4) primarily questioned the evaluation settings and the effectiveness of event information. We clarified that the ablation studies in Table 3 explicitly prove the significant contribution of event features . We also explained that our evaluation follows standard practices. Reviewer eG5r has acknowledged that our responses resolved their concerns regarding ablation and evaluation fairness, and has expressed a willingness to raise their score. Additionally, we responded to their follow-up question regarding how test view poses are obtained within the VGGT coordinate system.

---
- Reviewer RKQv (Initial Score: 6) found our idea of leveraging event cameras "novel and compelling" but noted a lack of evaluation on fast camera movements and questioned generalization to unseen trajectories . We addressed this by pointing to our existing experiments on the EvDeblurNeRF-DAVIS dataset and conducting additional experiments on the M3ED fast-motion dataset , showing that EA3D remains robust under rapid motion. We further elaborated on our trajectory augmentation strategy during training and referenced both our video demo and the experiments in Appendix B.3 to demonstrate robust generalization to test-time trajectories.

---
- Reviewer BEQ9 (Initial Score: 4) questioned the reliance on upstream MVS models (VGGT) and the novelty of incorporating 3D information into diffusion models. We clarified that relying on upstream MVS model (e.g., COLMAP) is standard practice in the field. We further highlighted the novelty in the implicit fusion of event geometry with RGB features via the EA-Renderer, which handles occlusion and fast motion more effectively than prior explicit RGB-only 3D guidance. We also updated the computation cost table to include upstream model runtimes as requested. Reviewer BEQ9 has confirmed that our clarifications on the upstream MVS model and novelty addressed their concerns and decided to raise their score.

---
- Reviewer osNE (Initial Score: 6) praised our "clear and robust architecture" and requested discussions on newer event-augmented Gaussian frameworks and more details on the geometry feature. We expanded the related work to include recent methods. We also provided additional visualizations (Figure 8) and detailed explanations of the geometry feature and our EA-Renderer to clarify the specific modeling insights that distinguish EA3D from prior work.

---

### Meta-Review · Area_Chair_2DH5 · 2026-01-08

**Summary:**

Reviewers generally receive the paper positively. Most positive reviewer (gMy4) is curious to see evaluation on real fast-motion scenes and comparisons with some other methods. Two most negative reviewers are eG5r and BEQ9. Reviewer eG54 concerns about a potentially unfair ablation. Reviewer BEQ9 concerns about the novelty of the method. Reviewer RKQv and osNE leaned positive at their initial reviews, but both request some additional evaluations or discussions.

**Reviewer Concerns:**

Most of the concerns by the reviewers seemed to address well by the authors.

**Reviewer Scores:**

I believe reviewer eG5r and BEQ9 will likely increase their scores as their comments indicate that the rebuttal address the concerns well. Other reviewers were already leaning positive might likely to keep their scores given not much lasting concerns.

---

### Decision · Program_Chairs · 2026-01-26

Accept (Poster)